# Specification of distinct cell types in a sensory-adhesive organ important for metamorphosis in tunicate larvae

Christopher J. Johnson[1], Florian Razy-Krajka[1], Fan Zeng[2], Katarzyna M. Piekarz[1], Shweta Biliya[3], Ute Rothbächer[2]*, Alberto Stolfi[1]*

**1** School of Biological Sciences, Georgia Institute of Technology, Atlanta, Georgia, United States of America, **2** Department of Zoology, University of Innsbruck, Innsbruck, Austria, **3** Molecular Evolution Core, Petit H. Parker Institute for Bioengineering and Bioscience, Georgia Institute of Technology, Atlanta, Georgia, United States of America

Ͽ These authors contributed equally to this work.
* ute.rothbaecher@uibk.ac.at (UR); alberto.stolfi@biosci.gatech.edu (AS)

**Data Availability Statement:** All relevant data are within the paper and its Supporting information

## Abstract

The papillae of tunicate larvae contribute sensory, adhesive, and metamorphosis-regulating functions that are crucial for the biphasic lifestyle of these marine, non-vertebrate chordates. We have identified additional molecular markers for at least 5 distinct cell types in the papillae of the model tunicate *Ciona*, allowing us to further study the development of these organs. Using tissue-specific CRISPR/Cas9-mediated mutagenesis and other molecular perturbations, we reveal the roles of key transcription factors and signaling pathways that are important for patterning the papilla territory into a highly organized array of different cell types and shapes. We further test the contributions of different transcription factors and cell types to the production of the adhesive glue that allows for larval attachment during settlement, and to the processes of tail retraction and body rotation during metamorphosis. With this study, we continue working towards connecting gene regulation to cellular functions that control the developmental transition between the motile larva and sessile adult of *Ciona*.

## Introduction

Tunicates, the sister group to the vertebrates, comprise a diverse group of marine non-vertebrate chordates [1,2]. Most tunicate species are classified in the order Ascidiacea, commonly known as ascidians [3], although phylogenetic evidence suggests this is not a monophyletic group within Tunicata [4–6]. The majority of ascidians have a biphasic life cycle that alternates between a swimming larva and a sessile adult. The larva functions exclusively to disperse the species, not feeding until it has found a suitable location on which to settle and trigger metamorphosis [7].

Recent work has started to reveal the cellular and molecular basis of larval settlement and metamorphosis. Key to the process of settlement and metamorphosis are the papillae, which comprise a set of 3 anterior sensory/adhesive organs in the laboratory model species of the

files, and from OSF: https://osf.io/wzrdk/ and https://osf.io/sc7pr/. Raw sequencing reads available from the SRA database under accession PRJNA949791.

**Funding:** This work was funded by National Science Foundation (NSF, www.nsf.gov) grant 1940743 and National Insititues of Health (www.nih.gov) grant GM143326 to AS; an NSF graduate fellowship to CJJ; and by Austrian Science Fund (FWF, www.fwf.ac.at) grant P 35402-B to UR. The funders did not play any role in study design, data collection and analysis, decision to publish, or preparation of the manuscript.

**Competing interests:** The authors have declared that no competing interests exist.

**Abbreviations:** ACC, axial columnar cell; CEN, caudal epidermal neuron; FP, fluorescent protein; IC, inner collocyte; OC, outer collocyte; PN, papilla neuron; PNA, peanut agglutinin; PSC, palp sensory cell; RTEN, rostral trunk epidermal neuron; scRNAseq, single-cell RNA sequencing; sgRNA, single-chain guide RNA; TEM, transmission electron microscopy.

genus *Ciona* and a majority of other ascidian genera as well (Fig 1) [8–11]. The papillae are composed of a few different cell types that have been characterized by both electron and fluorescence microscopy [9,12–14]. Several cells appear to secrete the "glue" or bioadhesive material required for the attachment of the larva to the substrate, termed "collocytes" [9,15]. Other cells are clearly neuronal (4 ciliated neurons per papilla) [9] and are required to trigger the onset of metamorphosis [16], which was also recently shown to depend on mechanical stimulation of the papillae [17]. Finally, at the very center of each papilla are 4 "Axial Columnar Cells" (ACCs), which have been suggested to possess chemosensory and contractile properties [11,18,19]. Although they have been called papilla "sensory cells" or "neurons," they are not innervated and have little structural and molecular overlap with the other 2 cell types. Furthermore, single-cell RNA sequencing (scRNAseq) revealed that they do not express genes typically associated with neuronal function [20].

In *Ciona*, previous work had established that the 3 papillae likely arise from 3 clusters of *Foxg+/Islet+* cells arranged roughly as a triangle—2 dorsal clusters (left and right) and single ventral cluster [21,22]. Although *Foxg* is initially activated in an entire row of cells at the very anterior of the neural plate, Sp6/7/8 (also known as Zfp220 or Buttonhead) is required to refine this swath of expression down to 3 "spots" of *Foxg*, which is required for expression of *Islet* in these cell clusters (Fig 1) [21]. MEK/ERK (e.g., MAPK) signaling also appears to play an important role in this refinement, as treatment with the MEK inhibitor U0126 results in a "U"-shaped band of *Islet* expression instead of 3 discrete foci (Fig 1) [22]. Similarly, BMP inhibition also causes a similar "U-shape" swath of *Foxg/Islet* expression, resulting in a single protrusion instead of the normal 3, termed the "*cyrano*" phenotype [23,24]. However, it has not been shown how these early specification events connect to the final cell type diversity and arrangement of the papillae.

Here, we describe novel genetic markers and reporter constructs that allowed us to visualize each of the different cell type of the papillae and follow their development upon various molecular perturbations targeting specific transcription factors or signaling pathways. We show that different transcription factors contribute to the specification of the different cell types and that cell-cell signaling in the FGF/MAPK and Delta/Notch pathways are crucial for patterning and arranging these cells in the 3 papillae. Altering papilla development in different ways contributes to different processes of post-settlement larval body plan rearrangements, revealing the complex molecular and cellular underpinning of tunicate larval metamorphosis.

## Methods

### Ciona handling

*Ciona robusta (intestinalis Type A)* were shipped from San Diego (M-REP), while *Ciona intestinalis (Type B)* were shipped from Roscoff Biological Station, France. Eggs were fertilized in vitro, dechorionated, and electroporated following established protocols [25–27]. Staging at different temperatures was estimated based on the published *C. robusta* developmental table from TUNICANATO [28]. Unc-76 tags were used as a default for fluorescent proteins (FPs) for optimal cell labeling as previously described [29], which excludes the FPs from the nucleus and ensures transport down axons. Typically, 40 to 100 μg of untagged or Unc-76-tagged FP plasmids and 10 to 35 μg of histone (H2B) fusion FP plasmids were used per 700 μl of electroporation solution. For CRISPR, typically 35 to 40 μg of Cas9 plasmid and 25 to 40 μg of each gRNA plasmid was used per 700 μl of electroporation solution, except when validating single-chain guide RNAs (sgRNAs) (see further below). For *Sp6/7/8*, *Pou4*, and *Foxg*, the 2 sgRNAs validated for each gene (S3 Fig) were used in combination. For *Villin*, all 3 validated sgRNAs were used in combination, while *Tuba3* only had 1 validated sgRNA. For *Islet*, most

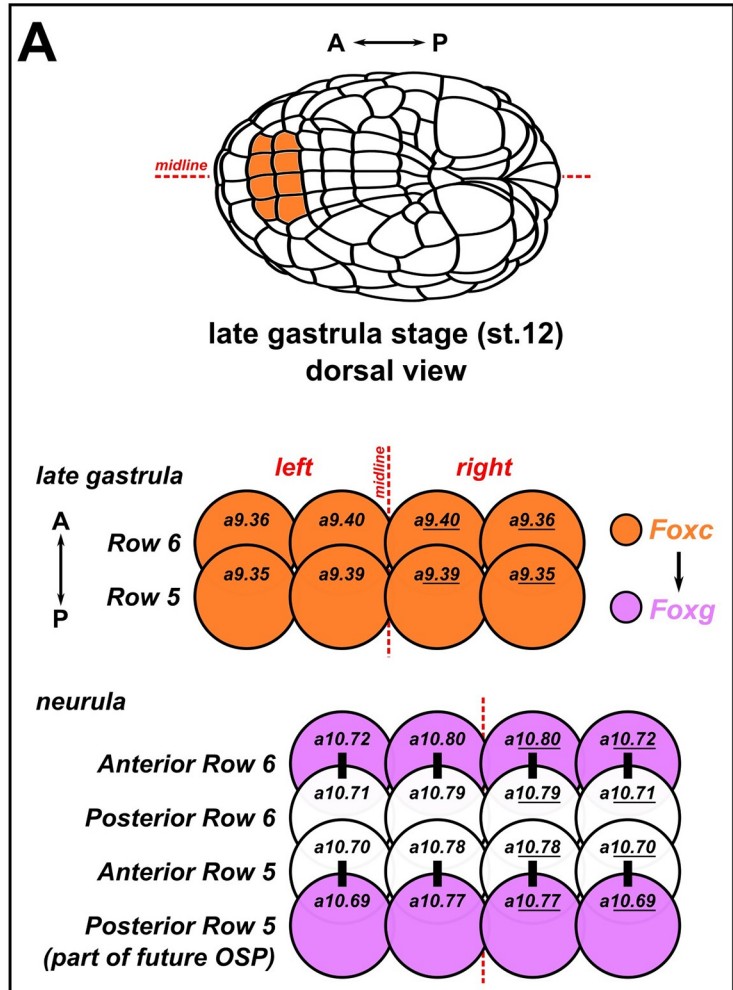
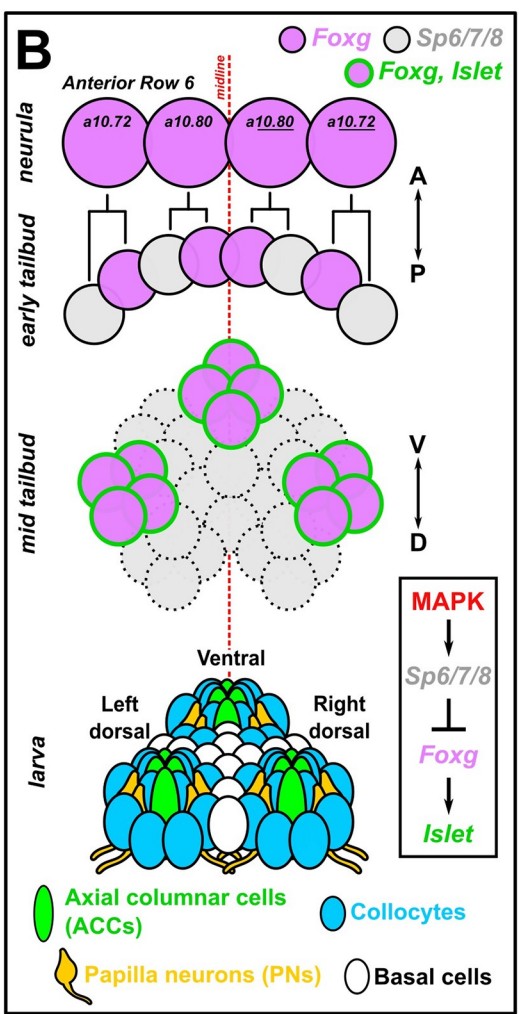

**Fig 1. Development of the papillae of Ciona.** (A) Diagram showing the early cell lineages that give rise to the papillae. The papillae invariantly derive from Foxc+ cells in the anterior neural plate, more specifically the anterior daughter cells of "Row 6" of the neural plate, which activate *Foxg* downstream of Foxc. *Foxg* is also activated in the posterior daughter cells of "Row 5," which go on to give rise to part of the OSP. Numbers in each cell indicate their invariant identity according to the Conklin cell lineage nomenclature. Black bars indicate sibling cells born from the same mother cell. (B) Diagram of what is currently known about the later lineage and fates of the *Foxg*+ "Anterior Row 6" cells shown in panel A. As the cells divide mediolaterally, some cells up-regulate *Sp6/7/8* and down-regulate *Foxg* (gray cells). Those cells that maintain *Foxg* expression turn on *Islet* and coalesce as 3 clusters of cells (pink with green outline): 1 medial, more ventral cluster, and 2 left/right, more dorsal clusters. Later, these 3 clusters organize the territory into the 3 protruding papillae of the larva, which contains several cell types described in detail by TEM [9]. Dashed cell outlines indicate uncertain number/provenance of cells. A-P: anterior-posterior. D-V: dorsal-ventral. Lineages and gene networks are based mostly on: [21,22,86,87]. OSP, oral siphon primordium; TEM, transmission electron microscopy.

experiments used only the *Islet.2* sgRNA, unless otherwise specified. Precise electroporation mixes for given perturbation experiments and controls are specified in the S1 File.

*C. robusta* embryos were raised at 20 ˚C and *C. intestinalis* embryos were raised at 18 ˚C, unless otherwise specified. For U0126 treatment, U0126 stock solution resuspended in DMSO was diluted to 10 μm final concentration in artificial seawater prior to transferring embryos at stage 16 (approximately 7.5 hpf). Negative control embryos were transferred to seawater with the equivalent volume of DMSO vehicle. For DMH1 treatment, concentrated stock solution was diluted to 2.5 μm final concentration in artificial seawater prior to transferring embryos at stage 10 (4 hpf), as previously established [23].

## Fixation, staining, imaging, scoring, and statistical analyses

Embryos and larvae were fixed for fluorescent protein imaging in MEM-FA fixation solution (3.7% formaldehyde, 0.1 M MOPS (pH 7.4), 0.5 M NaCl, 1 mM EGTA, 2 mM MgSO4, 0.1% Triton-X100), rinsed in 1× PBS, 0.4% Triton-X100, 50 mM NH4Cl, and 1× PBS, 0.1% Triton-X100. For mRNA in situ hybridization, embryos/larvae were fixed in MEM-PFA fixation solution (4% paraformaldehyde, 0.1 M MOPS (pH 7.4), 0.5 M NaCl, 1 mM EGTA, 2 mM MgSO4, 0.05% Tween-20) and in situ hybridization was carried out as previously described [30,31]. All probe template sequences are shown in the S1 File. Immunolabeling of Flag-tag (DYKDDDDK), β-galactosidase, mCherry (alone or in conjunction with mRNA in situ hybridization) was carried out as previously described [32], on embryos/larvae using mouse anti-DYKDDDDK Tag (Thermo Fisher catalog number MA1-91878, 1:1,000), mouse anti-β-gal (Promega catalog number Z3781, 1:1,000), and rabbit anti-mCherry (BioVision, accession number ACY24904, 1:500) primary antibodies. Specimens were mounted in 50% glycerol/1X PBS/2% DABCO mounting solution on slides with double-sided tape spacing between the slide and coverslip and imaged on Leica DM IL LED or DMI8 inverted epifluorescence microscopes, with maximum Z projection processing and cell measurements performed in LAS X.

PNA staining was carried out on 4% PFA fixed larvae, using Tris-buffered saline (pH 8.0) supplemented with 5 mM CaCl2 and 0.1% Triton X-100 (TBS-T). Unspecific background was blocked by 3% BSA in TBS-T for 2 h at room temperature. Biotinylated peanut agglutinin (PNA; Vector Laboratories, B-1075) was diluted in BSA-TBS-T to a final concentration of 25 μg/ml and applied to the specimen overnight at 4 ˚C. After several washes in TBS-T over 2 h, larvae were incubated for 1 h in fluorescent streptavidin (Vector Laboratories, SA-5006) diluted 1:300 in BSA-TBS-T at room temperature. PNA stainings were mounted in Vectashield (Vector Laboratories, H-1000-10) and imaged using a Leica SP5 II confocal scanning microscope. Stacks were acquired sequentially and z-projected. Images were analyzed with ImageJ (Version 1.52 h).

Only Foxc>H2B::mCherry+/lacZ+ embryos, larvae, and juveniles were scored, unless otherwise noted in results or figure and legend. For tests of proportion between 2 groups where there were 2 outcomes, Fisher's exact test was used, while for tests of proportion between more than 2 groups/outcomes, chi-square test was performed. All results of tests of proportions shown in S4 Data. For continuous variable measurements, see "Quantitative image analyses" subsection below.

## CRISPR/Cas9 sgRNA design and validation

The Cas9 [33] and Cas9::Geminin-Nterminus [34] protein-coding sequences have been described before. sgRNAs were designed using the CRISPOR website [35](crispor.tefor.net). Those sgRNAs with high Doench '16 score, high MIT specificity score, and not spanning known SNPs were selected for testing. Validation of sgRNAs was performed by co-electroporation 25 μg of *Eef1a>Cas9* or *Eef1a>Cas9::Geminin-Nterminus* and 75 μg of the sgRNA plasmid, per 700 μl of total electroporation volume. Genomic DNA was extracted from pooled larvae electroporated with a single sgRNA, using the QIAamp DNA micro kit (Qiagen). PCR products spanning each sgRNA target site were amplified from the corresponding genomic DNA, with primers designed so that the amplicon was to be 150 to 450 bp in size. Amplicons were purified by QIAquick PCR purification kit (Qiagen) and submitted for Amplicon-EZ Illumina-based sequencing by Azenta/Genewiz (New Jersey, United States of America), which returned mutagenesis rates and indel plots. CRISPR "rescue" cDNAs for *Islet*, *Foxg*, and *Sp6/7/8* were designed with silent (i.e., synonymous) point mutations disrupting our sgRNA targets sites and/or their PAMs (see S1 File). In the case of the *Islet.2* sgRNA, this one binds to a

sequence at an intron/exon boundary and therefore no mutation was needed for the rescue cDNA.

### RNA sequencing and analysis

The scRNAseq data from Cao and colleagues were re-analyzed in Seurat [36]. Combined larva stage data was clustered and plotted using 30 dimensions (S1A Fig). Clusters 3 and 33 were determined to contain papilla cell types and were re-clustered separately, also using 30 dimensions (S1B Fig). Differential gene expression plots (S1C Fig) were explored to find candidate papilla cell type markers, which appeared to be enriched in subclusters 8 and 9 (S1 Data). Some were then confirmed by in situ hybridization (S 1D Fig) and/or reporter plasmids. All code and Seurat files can be downloaded from: https://osf.io/sc7pr/. An alternative filtering and clustering approach (https://osf.io/dbv42) used in parallel to find specifically papilla neurons (PNs) resulted in a different TSNE plot (https://osf.io/6cg4h). From this, clusters 8, 10, and 18 were selected based on known papilla cell type markers and re-clustered, which led to the identification of a new subcluster "10" enriched for both ACC and PN markers. ACC markers (cluster "J") identified in Sharma and colleagues were subtracted from subcluster 10 markers to generate a list of potential PN-specific markers (https://osf.io/7xqp2).

Bulk RNA integrity numbers were determined using the Agilent Bioanalyzer RNA 6000 Nano kit and used as a QC measure. All samples with RINs over 7 were used for library preparation. mRNA was enriched using the NEBNext Poly(A) mRNA isolation module and Illumina compatible libraries were prepared using the NEBNext Ultra II RNA directional library preparation kit. QC on the libraries was performed on the Agilent Bioanalyzer 2100 and concentrations were determined fluorometrically. The libraries were then pooled and sequenced on the NovaSeq 6000 with an SP Flow Cell to get PE100bp reads.

The RNA-seq raw files were analyzed in Galaxy hub (usegalaxy.org) [37]. Firstly, the raw fastq files were inspected using FastQC Read Quality Reports (Galaxy Version 0.73+galaxy0) and MultiQC (Galaxy Version 1.11+galaxy0). The reads were then filtered and trimmed with Cutadapt (Galaxy Version 4.0+galaxy0). The minimum read length was set to 20 and the reads that did not meet the quality cutoff of 20 were discarded. Then, FastQC and MultiQC were used again to assess the resulting files after filtering and trimming. Next, the technical replicates were combined and used as the input to the mapping tool (RNA STAR, Galaxy Version 2.7.8a+galaxy0, length of the SA pre-indexing string of 12), together with the custom "KY21" version of the *Ciona* reference genome sequence and gene models ("Kyoto 2021", obtained from the Ghost Database; http://ghost.zool.kyoto-u.ac.jp/download_ht.html) [38]. The counts were generated using featureCounts (Galaxy Version 2.0.1+galaxy2; minimum mapping quality per gene was set to 10). Lastly, the differential gene expression analysis (S2 Data) was performed with DESeq2 (Galaxy Version 2.11.40.7+galaxy1). KY21 gene models were linked to KyotoHoya ("KH") version gene models using the Ciona Gene Model Converter application https://github.com/katarzynampiekarz/ciona_gene_model_converter. Raw sequencing reads available from the SRA database under accession PRJNA949791. Analysis code and files can be found at: https://osf.io/wzrdk/.

### Quantitative image analyses

Larvae subjected to papilla-specific knockout of *Islet*, *Villin*, or *Tuba3* (using *Foxc>Cas9*, see S1 File for detailed electroporation recipes) and negative control larvae were fixed at 17 hpf, 20 °C and mounted as above. *Islet intron 1 + -473/-9>Unc-76::GFP+* or *CryBG>Unc-76::GFP+* cells were imaged using a K3M camera mounted on a Leica DMI8 inverted epifluorescence microscope and the greatest distance between the apical and the basal extremities of each GFP

+ papilla (not individual cells) in LAS X, based on visible GFP fluorescence in the ACCs at a given focal plane (see example images with superimposed lines and measurements in S8D Fig). Sometimes, 2 or more papillae were GFP+ in the same larva. In these cases, each papilla was measured independently. Individual papilla length measurements are listed in S3 Data.

For analysis of Islet perturbations on *Villin* reporter expression, fluorescence from *Villin -1978/-1>Unc-76::GFP* and *Foxc>H2B::mCherry* reporters were acquired as above but with fixed illumination intensity and exposure times (50 ms for GFP, 100 ms for mCherry). Mean fluorescence intensities in both channels (GFP, mCherry) were measured in mCherry+ areas corresponding to the papillae, and ratio of mean values (GFP/mCherry) was calculated.

## Results

### Identification of novel markers and reporters for specific cell types in the papillae

We searched *Ciona robusta* (i.e., *intestinalis* Type A) whole-larva scRNAseq data [39] for evidence of the cell types described by transmission electron microscopy (TEM) of the papillae [9]. While a cell cluster annotated as "Palp Sensory Cells" (PSCs) appeared enriched for known markers of ACCs like *CryBG (KH.S605.3)* and *KH.C3.516* [20,40], genes expressed in other papilla cell types were also enriched in this cluster as well, including *Sp6/7/8 (KH.C13.22)* [21,22] and *Pou4 (KH.C2.42)* [16,23]. Re-analysis and re-clustering of these data revealed novel potential markers for different cell types in and around the papillae (S1A–S1C Fig and S1 Data). We performed in situ mRNA hybridization for several of these candidate markers in *C. robusta* larvae (S1D Fig). As we had hoped, they appeared to label different cells in the papilla territory. Some appeared to label cells in the center of each papilla, while others were expressed in cells surrounding or on the outermost edges of each papilla. These vastly different expression patterns supported the idea of mixed cell identities in the PSC scRNAseq cluster.

To further confirm the expression patterns of these and other candidate markers, we made reporter plasmids from their upstream *cis*-regulatory sequences and electroporated these into *Ciona* embryos. None of the selected genes showed any appreciable homology to genes of known function in other organisms, but we reasoned that they might serve as useful markers for specific papilla cell types. First, the gene *KH.L96.43*, predicted to encode a secreted protein with TSP1 repeats and a trypsin-like serine protease domain (S1E Fig and S1 File), was expressed in cells surrounding and in between the 3 papillae (S1D Fig). This pattern was recapitulated by a *KH.L96.43* reporter plasmid ("L96.43>GFP," Fig 2A). Co-electroporation with the papilla-specific *Foxg>mCherry* reporter [39] showed clear, mutually exclusive expression between the 2 reporters. We propose that L96.43 marks a population of "peri-papillary" and/or "inter-papillary" cells previously identified as "basal cells" that are part of the larger papilla region but excluded from the 3 protruding, *Foxg*+ papillae *sensu stricto* [9].

Next, we further confirmed that the PNs are distinct from the ACCs [9]. Previously identified as a potential PN marker by in situ hybridization [41], a *TGFB* reporter clearly labeled PNs (Fig 2B and S2A Fig), which are distinguished as the only papilla cell types bearing an axon [9]. However, co-electroporation of TGFB reporter with an ACC-specific *CryBG* reporter [40] appeared to result in "cross-talk," or cross-plasmid transvection in which a *cis*-regulatory element in one plasmid activates the transcription of a reporter protein-encoding gene on another, distinct co-electroporated plasmid (S2B Fig). Indeed, other PN-specific reporters tested did not cross-talk with *CryBG*, including the previously published *Gnrh1* reporter [42], and the novel reporter *KH.C4.78* ("C4.78>GFP") (S2C and S2D Fig). *KH.C4.78* encodes a predicted transmembrane protein with a single extracellular Sel1-like repeat (S1E Fig and S1 File). Interestingly, PN axons continued to extend posteriorly during the swimming phase to contact

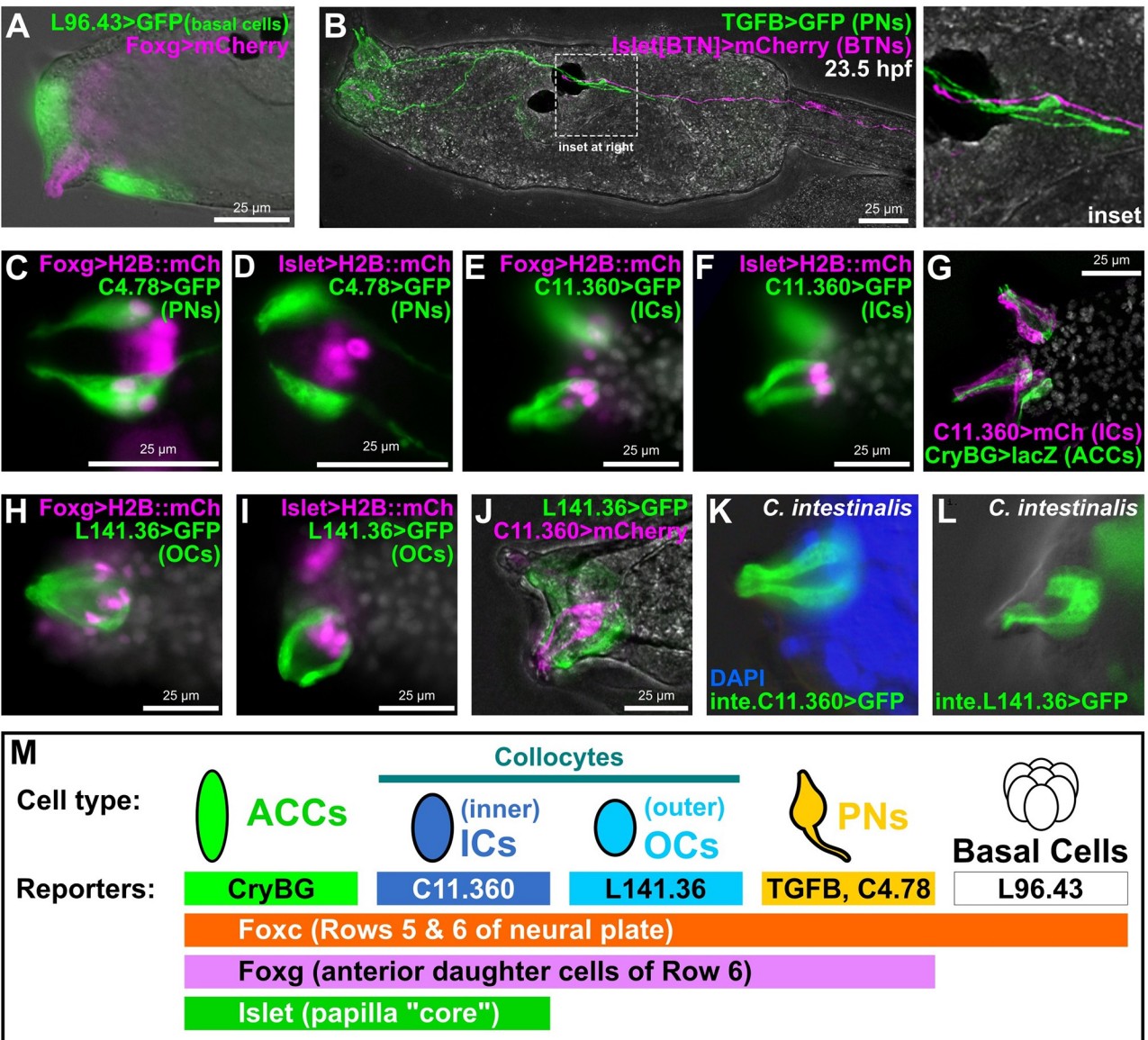

**Fig 2. Novel genetic markers label distinct cell types of the papillae.** (A) GFP reporter plasmid (green) constructed using the *cis*-regulatory sequences from the *KH.L96.43* gene labels basal cells in between and surrounding the protruding papillae labeled by *Foxg* reporter plasmid (pink). (B) *TGFB>GFP* reporter (green) labels PNs, the axons of which make contacts with BTN axons labeled by a BTN-specific *Islet* reporter (pink), at 23.5 h hpf, approximately corresponding to Hotta stage 30. (C) A *KH.C4.78* reporter (*C4.78>GFP*) also labels PNs, which are also labeled by *Foxg>H2B::mCherry (mCh)* reporter (pink nuclei). (D) Lack of overlap between expression of *C4.78>GFP* (green) and a papilla-specific *Islet* reporter plasmid (pink nuclei) showing that PNs do not arise from Islet+ cells. (E, F) Co-electroporation of *C11.360>GFP* (green) with H2B::mCherry reporter plasmids (pink nuclei) indicates these cells come from Foxg-expressing cells that also express Islet. (G) *C11.360>mCherry* reporter (pink) labels centrally located ICs adjacent to ACCs labeled by *CryBG>LacZ* reporter (green). (H, I) *L141.36>GFP* reporter (green) labels OCs that arise from *Foxg+* cells (pink nuclei) but do not express *Islet* (pink nuclei). (J) ICs and OCs are distinct cells as there is no overlap between *C11.360* (green) and *L141.36* (pink) reporter plasmid expression. (K) *Ciona intestinalis* (Type B) larva ICs labeled with a reporter plasmid made from the corresponding *cis*-regulatory sequence of the *C. intestinalis Chr11.1038* gene, orthologous to *C. robusta KH.C11.360*. (L) *C. intestinalis* larva OCs labeled by a *Chr7.130* reporter, corresponding to the *C. robusta* ortholog *KH.L141.36*. (M) Summary of the main marker genes and corresponding reporter plasmids used in this study to label different subsets of papilla progenitors and their derivative cell types. All GFP and mCherry reporters fused to the Unc-76 tag, unless specified (see Methods and supplement for details). Weaker *Foxg -2863/-3* promoter used in panel A, all other *Foxg* reporters used the improved *Foxg -2863/+54* sequence instead. All *Islet* reporters shown correspond to the *Islet intron 1 + bpFOG>H2B::mCherry* plasmid. White channel shows either DAPI (nuclei) and/or larva outline in brightfield, depending on the panel. All *C. robusta* raised at 20 °C to 18 hpf (roughly st. 28) except: panel B (23.5 hpf, ~st. 30); panels C–F (17 hpf, ~st. 27); panels H–J (20 hpf, ~st. 29). *C. intestinalis* raised at 18 °C to 20–22 hpf (Hotta stage 28). ACC, axial columnar cell; BTN, bipolar tail neuron; hpf, hours post-fertilization; IC, inner collocyte; OC, outer collocyte; PN, papilla neuron.

the anterior axon branches of the bipolar tail neurons (Fig 2B), which project their posterior axon branches to the very tip of the tail [43]. This hints at a potential mechanism for transducing sensory information from the papillae to the tail tip where tail retraction initiates, especially during later time points when larvae are competent to settle [44].

Double electroporation with *KH.C4.78* and *Foxg* reporters (Fig 2C and S2D Fig) revealed that, unlike the basal cells, PNs are specified from *Foxg*+ cells in the papillae. However, co-electroporation with a papilla-specific *Islet* reporter plasmid also revealed that PNs are adjacent to, but distinct from, the central Islet+ "core" of each papilla (Fig 2D and S2D Fig). In contrast, a *KH.C11.360* reporter ("*C11.360>GFP/mCherry*") labeled cells that were both Foxg+ and Islet +, but were clearly not the ACCs (Fig 2E–2G and S2C Fig). The *KH.C11.360* gene encodes a predicted secreted/transmembrane protein with no other recognizable domains or motifs (S1 File). The C11.360+ cells were adjacent to the ACCs but lacked the thin protrusions into the hyaline cap that are typical of the ACCs and also lacked axons typical of the PNs. Therefore, these cells appear to be collocytes, proposed to be adhesive-secreting cells responsible for attachment to the substrate during larval settlement [9].

Previous characterization of the papillae by TEM described 12 collocytes in each papilla [9], yet the C11.360 reporter appeared to only label at most 4 cells per papilla. This suggested the existence of cryptic collocyte subtypes. In fact, those same TEM images showed certain qualitative differences in cytoplasmic contents between peripheral collocytes and the more central collocytes [9]. Indeed, we identified another reporter, that of the gene *KH.L141.36* ("*L141.36>GFP*"), that labeled *Foxg*+ but *Islet*-negative cells that are at the periphery of each papilla but that are not PNs as they do not have axons (Fig 2H and 2I, and S2D Fig). *KH.L141.36* encodes a predicted transmembrane protein with at least 4 extracellular Sushi/SCR/CCP domains (S1E Fig and S1 File). Co-electroporation of *L141.36* and *C11.360* reporters labeled mutually exclusive groups of cells (Fig 2J and S2D Fig). We propose that these respective reporters delineate more peripheral, or "outer" collocytes (OCs) versus more central, or "inner" collocytes (ICs). Interestingly, strong *KH.L141.36* reporter expression was not visible in early larvae (approximately 17 hpf) like most of the other reporters described, suggesting a later onset of activation.

When using these *C. robusta* reporter plasmids to electroporate the closely related *C. intestinalis* (i.e., Type B) sourced from Roscoff, France [45], we noticed that their expression was very weak (S2E and S2F Fig). This led us to re-cloning the orthologous sequences from the *C. intestinalis* Type B genome [46] (S1 File). Percent identity over the alignable portions of these noncoding sequences (disregarding large gaps or insertions) was 89% for *C11.360* and 66% for *L141.36*. Electroporation of Type B embryos with Type B-specific reporter plasmids resulted in much stronger, reliable expression (Fig 2K and 2L). This suggests relatively significant changes to the *cis*-regulatory sequences of these cell type-specific genes in these otherwise nearly indistinguishable cryptic species.

Although we also obtained additional reporters that labeled 1 or more different papilla cell types (S2G and S2H Fig), we now had a full set of papilla cell type-specific marker genes and reporter plasmids for a deeper investigation of papilla patterning and development (Fig 2M). Finally, it is also important to note that some of these reporters are also expressed in cell types outside the papillae (e.g., *CryBG* in the otolith and *KH.C4.78* in the descending decussating neurons of the motor ganglion).

## Specification of ACCs, ICs, and OCs by Islet and Sp6/7/8 combinatorial logic

How are the cell types of the papillae (ACCs, ICs, OCs, and PNs) specified? In situ mRNA hybridization previously revealed partially overlapping expression territories of 3 genes

encoding sequence-specific transcription factors (Fig 3A): a central domain of *Islet+* cells, surrounded by a ring of cells that express both *Islet* and *Sp6/7/8* (and *Emx*, though distinct from the earlier expression of *Emx* at neurula stages), and additional cells surrounding them expressing only *Sp6/7/8* [22]. Additionally, overexpression of *Islet* had been previously shown to generate a single large papilla expressing the ACC reporter *CryBG>GFP* [22]. We therefore asked whether these transcription factors might be patterning the papillae into an ordered array of cell types (Fig 3A).

First we asked, does Islet specify the centrally located ACCs and ICs? To test this, we turned to tissue-specific CRISPR/Cas9-mediated mutagenesis [33]. To knock out *Islet* in the papillae, we electroporated a previously validated sgRNA expression construct targeting its intron/exon 2 boundary (*U6>Islet.2*, 44% mutagenesis efficacy, S3 Fig) [47] together with *Foxc>Cas9*. Papilla-specific CRISPR-based knockout of *Islet* and resulting loss of ACC cell fate was confirmed by loss of *CryBG>GFP* expression, compared to negative control individuals electroporated instead with previously published *U6>Control* sgRNA vector [33] targeting no sequence (Fig 3B and 3C, and S4A Fig). *CryBG>GFP* activation was rescued by expressing an *Islet* cDNA that is not targeted by our sgRNAs, demonstrating that its loss is not likely due to off-target CRISPR effects (S5A Fig). Therefore, we conclude that *Islet* is required for the specification and differentiation of ACCs. A smaller portion of larvae completely lost expression of the IC reporter, *C11.360>GFP*, but expression was still substantially reduced relative to the control (Fig 3B and 3C). This difference might be due to lower sensitivity of the IC reporter to *Islet* knockout, or might simply reflect the lower level of mosaicism of *C11.360>GFP* expression observed in the control.

To test whether *Islet* is required for other cell types of the papillae, we repeated papilla-specific *Islet* CRISPR knockout using our different reporters to monitor the specification or differentiation of OCs *(L141.36>GFP)* and PNs *(TGFB>GFP)*. While *Islet* knockout altered the general morphology of the papillae (see further below), it did not cause significant loss of OC or PN reporter expression (Fig 3B and 3C, and S4A Fig). We therefore conclude that *Islet* is required for the specification and/or differentiation of ACCs and ICs, but not OCs or PNs.

Because it was reported that an outer *Emx+* "ring" of *Islet+* cells in each papilla co-express *Sp6/7/8* [22], we hypothesized that *Sp6/7/8* might be required for a fate choice between ACCs and ICs. Corroborating the idea that these outer *Islet+* cells are specified as ICs, we cloned an intronic *cis*-regulatory element from the *Emx* gene that is sufficient to drive late expression specifically in ICs (S2G Fig). This late ring of *Emx* expression is not to be confused with the earlier expression of *Emx* in *Foxc+/Foxg*-negative cells at the neurula stage [21], which represent a distinct lineage (Fig 1). To test the role of Sp6/7/8 in IC versus ACC fate choice, we used the papilla-specific *Islet cis*-regulatory element to overexpress Islet or Sp6/7/8. While *Islet>Islet* did not reduce expression of either reporter (Fig 4A and 4B, with overexpression confirmed by immunostaining for a Flag tag epitope fused to Islet, S2I Fig), *Islet>Sp6/7/8* specifically abolished ACC reporter expression, but not that of the IC reporter (Fig 4A and 4B). In fact, IC reporter expression appeared to be slightly expanded in approximately 29% of larvae electroporated with *Islet>Sp6/7/8*, as assayed by perfect overlap with *Islet>H2B::mCherry* reporter (S5C Fig). Taken together, these results suggest that overexpression of Sp6/7/8 in the Islet + cells of the papillae is sufficient to abolish ACC fate and might convert the cells to an IC fate instead.

To further show that the combination of Islet and Sp6/7/8 is sufficient to specify IC cell fate, we used the *Foxc* promoter to drive expression of Islet, Sp6/7/8, or a combination of both in the entire papilla territory. *Foxc>Islet* alone strongly promoted ACC reporter expression, as previously reported [22], but resulted in more scattered IC reporter expression (Fig 4C and 4D). In contrast, co-electroporation of *Foxc>Islet* and *Foxc>Sp6/7/8* resulted more often in a

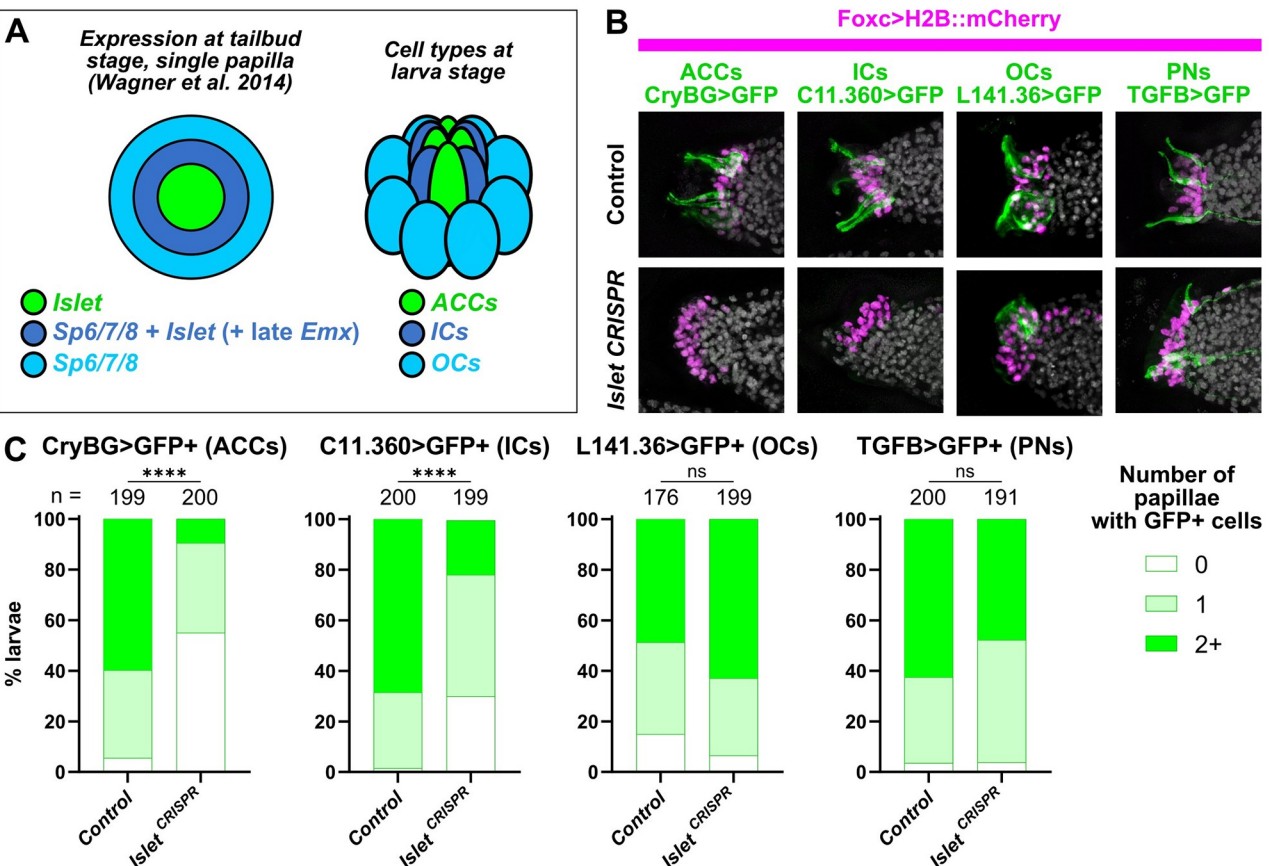

**Fig 3. The transcription factor Islet is required for specification of ACCs and ICs.** (A) Diagram depicting a partially overlapping expression patterns of *Islet* and *Sp6/7/8*, as originally shown by in situ mRNA hybridizations (Wagner and colleagues), and the correlation of these patterns with the later arrangement of ACCs, ICs, and OCs in the papillae. "Late" *Emx* expression in a ring of cells expressing both *Islet* and *Sp6/7/8* appears to be distinct from earlier *Emx* expression in *Foxg-negative* cells (see text and S2 Fig for details). (B) Papilla lineage-specific CRISPR/Cas9-mediated mutagenesis of *Islet* using *Foxc>Cas9* and a the *U6>Islet.2* sgRNA plasmid shows reduction of larvae showing expression of reporters labeling ACCs and ICs, but not OCs or PNs. Results compared to a negative "control" condition using a negative control sgRNA (*U6>Control*, see text for details). Nuclei counterstained with DAPI (white). (C) Scoring data for larvae represented in panel B, averaged between biological duplicates. *Foxc>H2B::mCherry+* larvae were scored for quantity of papillae showing visible expression of the corresponding GFP reporter plasmid. Due to mosaic uptake or retention of the plasmids after electroporation, number of papillae with GFP fluorescence is variable and rarely seen in all 3 papillae even in control larvae. Normally larvae have 3 papilla (GFP+ or not), but some mutants have more/fewer than 3. ACC/IC/OC subpanels in panel B at 20 hpf/20 ˚C (~st. 29), PN subpanels at 21 hpf/20 ˚C (~st. 29). Same applies to scoring data in Panel C. All experiments were performed in duplicate, with number of embryos ranging from 76 to 100 per condition per replicate. \*\*\*\* *p* < 0.0001 in both duplicates, as determined by chi-square test. ns = not statistically significant in at least 1 duplicate, also by chi-square test. See S4 Data for the data underlying the graphs and for statistical test details. ACC, axial columnar cell; IC, inner collocyte; OC, outer collocyte; PN, papilla neuron; sgRNA, single-chain guide RNA.

large, single papilla expressing predominantly the IC reporter, not the ACC reporter (Fig 4C and 4D).

Finally, we performed papilla-specific CRISPR knockout of *Sp6/7/8*, following the same strategy for *Islet* detailed above, using a combination of 2 new sgRNAs that were designed and validated (S3A Fig). Indeed, CRISPR/Cas9-mediated mutagenesis of *Sp6/7/8* in the papilla territory resulted in loss of IC cell fate, as assayed by expression of *C11.360>GFP* (Fig 4C and 4E). Reduced IC reporter expression was also observed using either individual *Sp6/7/8* sgRNAs independently (S4B Fig), and was overcome by *Foxc* promoter-driven overexpression of an *Sp6/7/8* rescue cDNA (S5B Fig), suggesting this effect was not due to CRISPR off-targeting. In

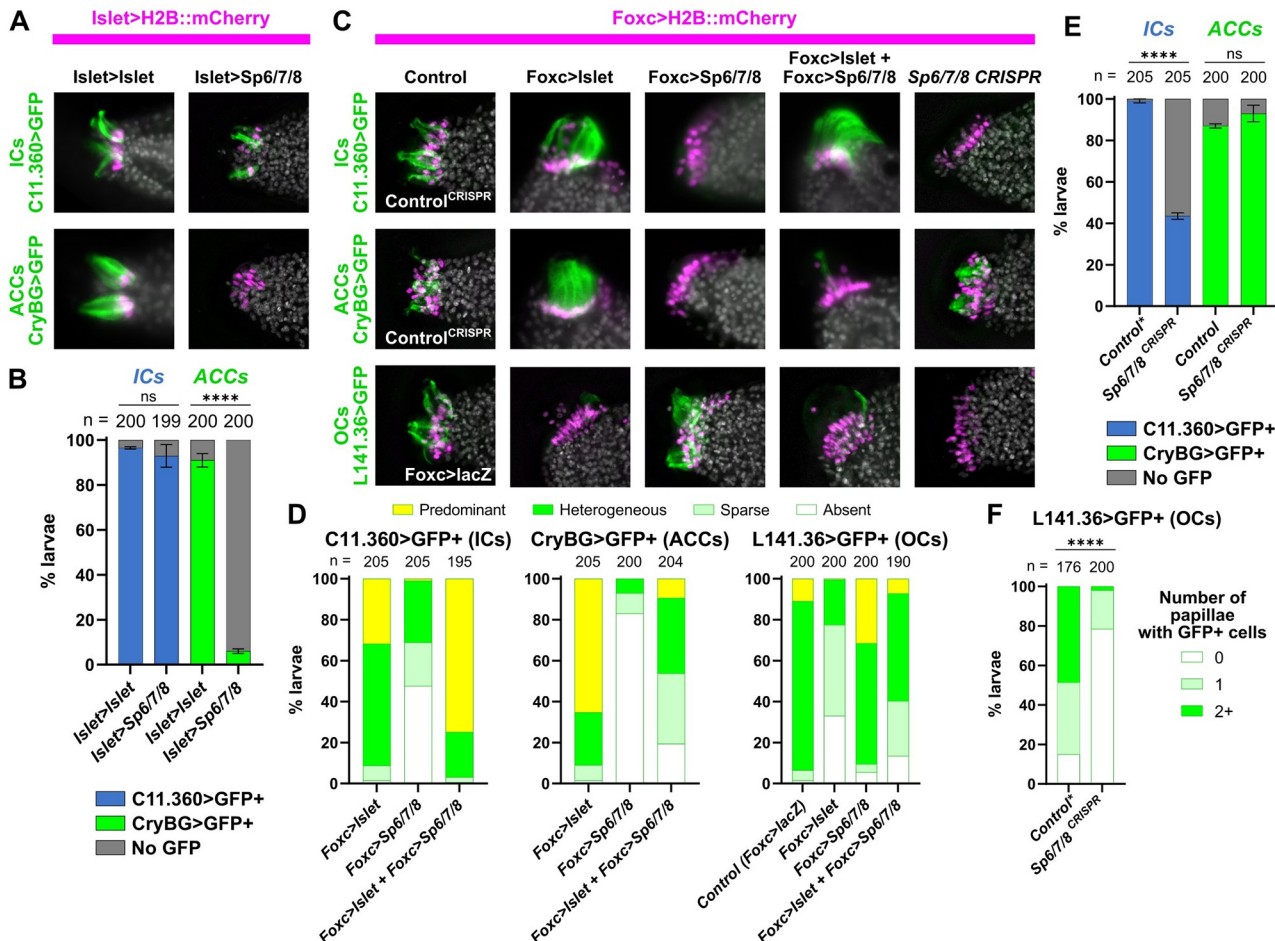

**Fig 4. Specification of ACCs, ICs, and OCs by a combinatorial logic of Islet and Sp6/7/8.** (A) Overexpression of Sp6/7/8 (using the *Islet>Sp6/7/8* plasmid) in all Islet+ papilla cells results in loss of ACCs (assayed by expression of *CryBG>Unc-76::GFP*, green), but not of ICs (assayed by expression of *C11.360>Unc-76::GFP*, green). Islet overexpression (with *Islet>Flag::Islet-rescue*) does not significantly impact the specification of ACCs or ICs. Larvae at 20 hpf/20 °C (~st. 29). (B) Scoring data showing presence or absence of ICs or ACCs in *Foxc>H2B::mCherry+* larvae, as represented in panel A. Experiments were performed in duplicate with 99 or 100 larvae in each duplicate. (C) Cell type specification assayed by reporter plasmid expression (green) in larvae subjected to various *Islet* and/or *Sp6/7/8* perturbation conditions (see main text for details). For ICs and ACCs, the "control" condition is negative control CRISPR (*U6>Control*), while for OCs it is *Foxc>lacZ*. Overexpression ACC/IC subpanels are at 18.5 hpf/20 °C (~st. 28), all CRISPR and OC panels at 20 hpf/20 °C (~st. 29). (D) Scoring data for most larvae represented in panel C. Foxc>H2B::mCherry+ larvae were scored for cell type-specific GFP reporter expression that was "heterogeneous" (mixed on/off GFP expression, with all "wild type" patterns of expression falling under this category), "predominant" (ectopic/supernumerary GFP+ cells), "sparse" (reduced frequency/intensity of GFP expression), or "absent" (no GFP visible). (E) IC or ACC reporter (*C11.360>Unc-76::GFP*, *CryBG>Unc-76::GFP*) expression scored in *Foxc>H2B::mCherry+* larvae represented in top 2 panels of right-most column in C. Experiment was performed and scored in duplicate, with number of larvae in each duplicate ranging from 100 to 105. (F) OC-specific reporter (*L141.36>Unc-76::GFP*) expression scored in *Foxc>H2B::mCherry+* larvae represented by the bottom/right-most subpanel in panel B. Scoring strategy same as in Fig 3. Asterisk denotes when a duplicate of the negative control condition was also used for plots in Fig 3, as multiple CRISPR experiments were performed in parallel. All experiments were performed in duplicate, with number of embryos ranging from 76 to 100 per duplicate. *Foxc>Cas9* used for all CRISPR/Cas9 experiments. The *Islet cis*-regulatory sequence used (panels A and B) was always *Islet intron 1 + -473/-9*. For overexpression conditions, *Foxc>lacZ* or *Islet>LacZ* were used to normalize total amount of DNA (see S1 File for detailed electroporation recipes). All error bars indicate upper and lower limits. **** *p* < 0.0001 in both duplicates as determined by Fisher's exact test (panels B and E) or chi-square test (panel F). ns = not statistically significant in at least 1 duplicate. See S4 Data for the data underlying the graphs and statistical test details. ACC, axial columnar cell; IC, inner collocyte; OC, outer collocyte.

contrast, the same perturbation did not diminish the expression of the ACC reporter (Fig 4C and 4E).

We noticed that *Foxc>Sp6/7/8* alone resulted in a large proportion of larvae lacking either ACC or IC reporter expression (Fig 4D). This suggested the possibility that Sp6/7/8 alone

might be promoting another papilla cell fate. Indeed, we found that *Sp6/7/8* knockout by CRISPR abolishes the expression of the OC reporter (*L141.36>GFP*), while *Foxc>Sp6/7/8* expands it slightly (Fig 4C, 4D and 4F). In contrast, *Foxc>Islet* alone or in combination with *Foxc>Sp6/7/8* suppressed OC reporter expression (Fig 4C and 4D), while *Islet* knockout did not affect it, as shown further above (Fig 3B and 3C, and S4A Fig). Taken together, these results suggest that a combinatorial transcriptional logic underlies papilla cell fate choices between ACCs (Islet alone), ICs (Islet + Sp6/7/8), and OCs (Sp6/7/8 alone).

## Identifying the adhesive-secreting cells of the papillae

Previous data revealed PNA staining as a marker for glue-secreting cell granules, the adhesive papillary cap, and adhesive prints left by larvae on the substrate [9,15]. The delineation of 2 collocyte populations opened the question of whether both (ICs and OCs) are equally PNA-positive. To answer this question, we performed PNA stainings on larvae expressing IC or OC reporter plasmids (Fig 5A and 5B). Interestingly, ICs contained PNA-stained granules only at

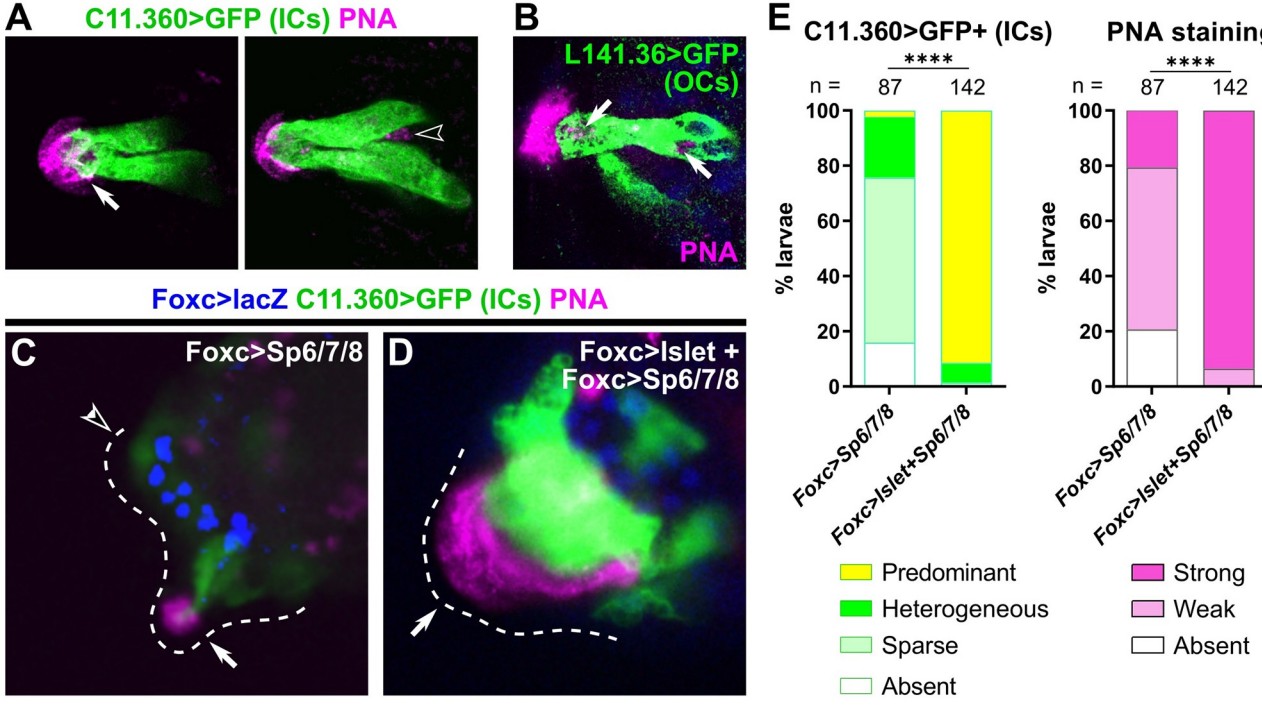

**Fig 5. Both types of collocytes contribute to production of adhesive material.** (A) PNA-stained granules (pink) are seen in the hyaline cap and the apical tip of ICs (left panel, white arrow) in a *C. intestinalis* larva labeled by the *C. intestinalis C11.360>Unc-76::GFP* reporter (green). PNA-stained granules are also seen in cells not labeled by the IC reporter (right subpanel, hollow arrowhead), suggesting they are localized in a different cell type. Left and right subpanels are from different focal planes of the same papilla. (B) OCs labeled with *C. robusta L141.36>Unc-76::GFP* (green) in a *C. robusta* larva, with PNA-stained granules (pink) in both apical and basal positions within the cell (white arrows). DAPI in blue. (C) PNA staining (pink) in *C. robusta* upon overexpression of *Sp6/7/8* alone, showing reduction of IC specification as assayed by *C11.360>Unc-76::GFP* expression (green). Weak PNA staining and GFP expression are still visible in some papillae (solid arrow), but not others (open arrowhead). (D) PNA staining (pink) and *C11.360>Unc-76::GFP* expression (green) in *C. robusta* upon overexpression of both *Islet* and *Sp6/7/8*, showing expansion of IC fate in a single large papilla (arrow). PNA staining is similarly expanded over the entire IC cluster, confirming that ICs produce the adhesive glue. *Foxc>lacZ* expression (β-galactosidase immunostaining) shown in blue in both C and D. (E) Scoring of larvae represented in panels C and D, averaged across duplicates. Weak PNA staining is observed upon partial suppression of IC fate, but strong PNA staining is seen upon expansion of supernumerary ICs, confirming that this cell type is one of the major contributors of PNA-positive adhesive glue. Total larvae (duplicate 1) or β-galactosidase+ larvae (duplicate 2) were scored. **** *p* < 0.0001 in both duplicates, as determined by chi-square test. See S4 Data for sample size, statistical test details, and for the data underlying the graphs. *C. intestinalis* raised to 20–22 hpf at 18 ˚C (~st. 28), *C. robusta* raised to 20 hpf at 20 ˚C (~st. 29). See Supplemental Movies for full confocal stacks and S6 Fig for single-channel images. IC, inner collocyte; OC, outer collocyte; PNA, peanut agglutinin.

the very apical tip, on top of which the strongest PNA staining is seen extracellularly (Fig 5A and S6 Fig and S1 Movie), while the majority of PNA-stained intracellular granules were not within the ICs at this stage (Fig 5A). Consistently, the OCs were the main cells showing PNA-stained granules located within the papillae (Fig 5B and S6 Fig and S2 Movie). This distribution of PNA staining corresponds to the distribution of granules previously identified by high-pressure freezing electron microscopy [9], in which collocytes located in the central core of the papilla contain granules mostly at their apical end. Indeed, in cross-sections, granules were most abundant inside the papillary body, likely in cells identified here as OCs.

To further investigate the contributions of both ICs and OCs to glue secretion, we performed PNA staining on larvae in distinct perturbation conditions. Namely, we electroporated larvae with *Foxc>Sp6/7/8*, which was shown above to suppress IC specification, or with *Foxc>Islet* and *Foxc>Sp6/7/8* combined, which was shown to convert most of the papilla territory into ICs. Although *Foxc>Sp6/7/8* eliminated most IC reporter expression, PNA staining was still weakly present (Fig 5C and 5E), likely due to continued presence of OCs. In contrast, *Foxc>Islet + Foxc>Sp6/7/8* resulted in a single enlarged papilla with supernumerary ICs, and the entire papilla was often covered by strong PNA staining (Fig 5D and 5E). Taken together, these results suggest that both ICs and OCs contribute to the production of adhesive material, but that the ICs (or their progenitors) are likely the more important contributors.

## Specification of PNs and OCs from cells that have down-regulated Foxg

With the specification of ACCs/ICs/OCs explained in large part due to overlapping expression domains of Islet and Sp6/7/8, the precise developmental origins of the PNs and OCs still remained elusive. While it has become clear that the Islet+ cells at the core of each papilla give rise to ACCs and ICs, Papilla-specific *CRISPR* knockout of *Islet* did not abolish PNs or OCs, as shown above (Fig 3). This suggested they do not arise from these core Islet+ cells, consistent with their more lateral positions as shown previously by TEM [9]. Furthermore, recently published in situ hybridization data showing presumptive *Pou4*-expressing PN precursors surrounding *Islet*-expressing cells at late tailbud stage [23]. Indeed, co-electroporation of *Islet* reporter and PN- or OC-specific reporter plasmids clearly showed PNs and OCs immediately adjacent to, but distinct from, Islet+ cells (Fig 2D and 2I, and S2D Fig).

Might PNs and OCs be arising from the cells in which *Foxg* is down-regulated (likely via repression by Sp6/7/8) and that do not go on to express *Islet* (Fig 6A) [21–23]? To test this, we used the MEK (MAPK kinase) inhibitor U0126 to expand *Islet* expression as previously done (Fig 6A) [22]. While treatment with 10 μm U0126 at 7.5 hpf (between stages 16 and 17, or late neurula and early tailbud) predictably expanded *Islet* reporter expression, it also eliminated expression of the PN reporter *C4.78>GFP*, as well as that of the OC reporter *L141.36>GFP* (Fig 6B and 6C). These results suggest that *Foxg+* papilla cells that maintain *Foxg* expression go on to express *Islet* and give rise to ACCs and ICs, while the cells that activate *Sp6/7/8* and down-regulate *Foxg* in response to MAPK signaling go on to give rise to OCs and PNs instead.

## PNs are specified by common peripheral neuron regulators

Previous papilla-specific TALEN knockout of the neuronal transcription factor-encoding gene *Pou4* successfully eliminated PNs and the larva's tail resorption response to mechanical stimuli [16]. Pou4 has been previously implicated in a Myt1-dependent regulatory cascade that specifies the caudal epidermal neurons (CENs) of the tail, from neurogenic midline cells expressing the proneural bHLH transcription factor *Ascl.a* (*KH.L9.13*, sometimes called *Ascl2* or *Ascl.b* previously) [23,48–51]. To precisely visualize the neurogenic cells of the papillae, we performed double (two-color) mRNA in situ hybridization for *Ascl.a* and *Foxg* at the mid-tailbud

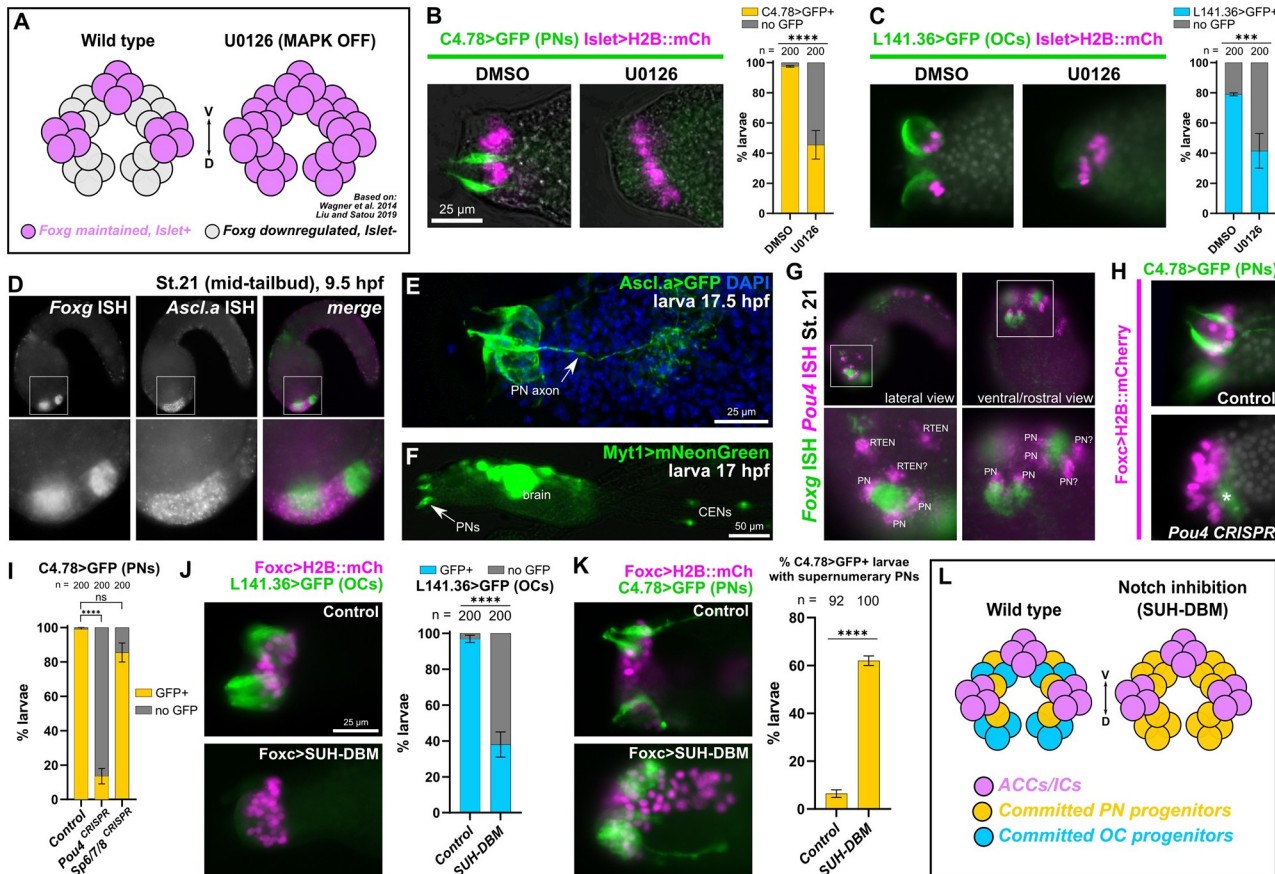

**Fig 6. Specification of PNs and OCs from *Islet*-negative cells by MAPK and Notch pathways.** (A) Diagram showing effect of MAPK inhibition with the pharmacological MEK inhibitor U0126, based on findings from Wagner and colleagues and Liu and Satou. Inhibition of FGF/MAPK results in expansion of *Foxg* and *Islet* from 3 discrete foci to a large "U-shaped" swath, transforming 3 papillae into a single, enlarged papilla (similar results reported with BMP inhibition by Roure and colleagues). (B) The 10 μm U0126 treatment at 7.5 hpf/20 ˚C (st. 16) results in loss of PNs (assayed by *C4.78>Unc-76::GFP* at 17 hpf/20 ˚C, ~st. 27, green) upon expansion of Islet+ cells (pink nuclei), relative to DMSO alone. (C) The same treatment results in loss of OCs (*L141.36>Unc-76::GFP* at either 17 or 20 hpf/20 ˚C, ~st. 27–29 green) upon expansion of Islet+ cells (pink nuclei). Both U0126 experiments were performed in duplicate, with 100 larvae per condition per duplicate. (D) Two-color, whole-mount mRNA in situ hybridization for *Foxg* (green in merged image) and *Ascl.a* (*KH. L9.13*, pink). (E) Larva electroporated with *Ascl.a>Unc-76::GFP* labeling several papilla cells including PNs. (F) *Myt1>mNeonGreen* labeling PNs and other other neurons including CENs. (G) Two-color in situ hybridization of *Foxg* (green) and *Pou4* (pink), the latter labeling adjacent PNs and possibly RTENs. (H) Lineage-specific CRISPR/Cas9-mediated mutagenesis of *Pou4* results in loss of PN reporter expression *(C4.78>Unc-76::GFP*, green). Asterisk denoted background/leaky expression in mesenchyme. Larvae at 17 hpf/20 ˚C (~st. 27). (I) Scoring of *Foxc>H2B::mCherry+* larvae represented in panel H and in *Sp6/7/8* CRISPR mutagenesis condition showing Sp6/7/8 does not appear to play a major role in PN reporter expression like Pou4. Experiments repeated in duplicate with 100 larvae in each. (J) Inhibition of Delta/Notch signaling using *Foxc>SUH-DBM* results in reduced expression of OC reporter (*L141.36>Unc-76::GFP*, green) at 21 hpf/20 ˚C (~st. 29). Experiment was repeated in duplicate with 100 larvae in each. (K) Notch inhibition also results in concomitant expansion of supernumerary PNs at 17 hpf/20 ˚C (~st. 27, labeled by *C4.78>Unc-76::GFP*, green) relative to *Foxc>lacZ* control. Experiment was repeated in duplicate, with 42 to 50 larvae in each. (L) Summary diagram and model of effects of Delta/Notch inhibition on PN/OC fate choice in *Islet*-negative (but formerly *Foxg+*) papilla progenitor cells. All *Islet* reporters are the *Islet intron 1 + bpFOG>H2B::mCherry*. All error bars indicate upper and lower limits. **** $p < 0.0001$, *** $p = 0.0003$ in both duplicates as determined by Fisher's exact test, ns = not statistically significant in at least 1 duplicate. See S4 Data for the data underlying the graphs and for statistical test details. CEN, caudal epidermal neuron; OC, outer collocyte; PN, papilla neuron; RTEN, rostral trunk epidermal neuron.

stage. Indeed, *Ascl.a* expression was seen broadly in the papilla territory surrounding the 3 *Foxg+* cell clusters (Fig 6D). This was confirmed by an *Ascl.a* fluorescent protein reporter plasmid that labeled a broad set of papilla territory cells, including PNs and their axons (Fig 6E). Furthermore, a previously published *Myt1* reporter [52] was also found to be expressed in the PNs (Fig 6F). Double in situ of *Pou4* and *Foxg* revealed *Pou4+* cells surrounding each *Foxg+*

cluster, corroborating a recent report [23] (Fig 6G). It was not immediately clear which Pou4 + cells were PN precursors and which were nearby rostral trunk epidermal neuron (RTEN) precursors. Based on our images and those of the most recent study [23], we propose that there are initially 2 Pou4+ cells per papilla, later dividing to give rise to the 4 PNs per papilla as previously described [9]. This would mirror the development of the epidermal neurons of the tail, in which neurons are born side-by-side as pairs after a final cell division by a committed mother cell [49]. Papilla-specific CRISPR knockout of *Pou4* with a combination of 2 newly validated sgRNAs (S3D Fig) recapitulated the loss of PN differentiation by the previously published TALEN knockout [16], as assayed by *C4.78* and *TGFB* reporter expression (Fig 6H and 6I, and S4 Fig). In contrast, *Pou4* knockout had no effect on the specification of ACCs or OCs, suggesting Pou4 function is specific for PN fate in the papillae (S4 Fig). Taken together, these results suggest that PNs are specified from interspersed neurogenic progenitors that are carved out by MAPK signaling. Interestingly, CRISPR knockout of *Sp6/7/8* did not substantially affect PN specification (Fig 6I). This suggests that even though Sp6/7/8 down-regulates *Foxg* in these cells [21], it does not appear to be required for their neurogenic potential.

## Notch signaling regulates the fate choice between PNs and OCs

Because both OCs and PNs appeared to arise from *Foxg*-down-regulating, *Islet*-negative cells, we sought to test whether an additional regulatory step is required for the fate choice between these 2 cell types. In the neurogenic midline territory of the tail epidermis, lateral inhibition by Delta/Notch signaling regulates the final number and spacing of CENs [48,49,53]. Delta/Notch limits the expression of Myt1, which in turn activates *Pou4* expression. In the tail epidermis, the major ligand involved is the putative Delta like non-canonical Notch ligand homolog (encoded by gene *KH.L50.6)*, which is also expressed in alternating pattern in the papillae (S7A Fig). We therefore decided to test whether a similar mechanism in controlling the number of PNs and OCs surrounding each papilla. To test the requirement of Delta/Notch, we overexpressed a DNA-binding mutant of the Notch co-factor RBPJ/SUH (SUH-DBM) [54]. Indeed, electroporation with *Foxc>SUH-DBM* resulted in loss of OC reporter expression (Fig 6J), and concomitant expansion of PN reporter expression (Fig 6K). We conclude that Delta/Notch signaling regulates PN versus OC fate choice in neurogenic progenitor cells surrounding each presumptive papilla, with Notch delimiting the specification of supernumerary neurons, thus allowing OCs to form (Fig 6L). In contrast, SUH-DBM had minimal effect on IC/ACC fate choice (S7B Fig), suggesting Delta/Notch might only regulate neurogenesis, and not cell fate choice in general, in the papilla territory.

This common origin of PNs and OCs is also supported by the recent finding that the latter appear to have basal bodies like the PNs, but without the accompanying sensory cilia [9]. Interestingly, papilla-specific knockout of *Foxg* resulted in moderate loss of PN reporter expression (*TGFB>GFP)*, and very little effect on the OC reporter (S4A Fig). This suggests differing requirement for Foxg in different cell type-specific branches of the papilla regulatory network, despite all these cell types arising from cells that initially express Foxg.

## Regulation of papilla morphogenesis by Islet

It was previously shown that *Foxg* or *Islet* overexpression induces the formation of a single enlarged "megapapilla," in which all cells are substantially elongated relative to the rest of the epidermis [21,22]. We have shown above that this appears to be driven by expansion of ACCs and/or ICs, which are atypically elongated in the apical-basal direction and form apical protrusions and microvilli. Islet is sufficient for apical-basal elongation of epidermal cells [22], and morpholino-knockdown of *Foxg* (which is upstream of *Islet)* also impairs proper papilla

morphogenesis [21]. We asked if *Islet* is required for papilla morphogenesis, using papilla-specific CRISPR knockout of *Islet*. Knocking out *Islet* in the papilla territory impaired the formation of the typically "pointy-shaped" papillae, resulting instead in blunt cells with flat, broader apical surfaces and reduced cell length along the apical-basal axis (Fig 7A and 7B, S8C and S8D Fig). This result suggested that transcriptional targets downstream of *Islet* might be regulating the distinct cell shape of ACCs/ICs.

To identify potential candidate effectors of morphogenesis downstream of *Islet*, we used bulk RNAseq to measure differential gene expression between different Islet perturbation conditions. We compared "negative control" embryos to (1) embryos in which *Islet* was overexpressed in the whole territory using the *Foxc* promoter (*Foxc>Islet*); and (2) embryos in which *Islet* was knocked out specifically in the papilla lineage by CRISPR/Cas9. For this, we designed an additional sgRNA targeting the first exon of *Islet*, to be used in combination with the already published sgRNA to generate larger deletions. This new sgRNA vector, which we named *U6>Islet.1*, resulted in a mutagenesis efficacy of 20% (S3B Fig).

Whole embryos from each condition were collected at 12 hpf (*Islet* conditions) at 20 ˚C in biological triplicate. RNA was extracted from pooled embryos in each sample, and RNAseq libraries were prepared from poly(A)-selected RNAs and sequenced by Illumina NovaSeq. This bulk RNAseq approach revealed that Islet overexpression results in the up-regulation of several ACC markers from previous scRNAseq analysis (S2 Data) [20]. With Islet overexpression, this included ACC markers previously validated by mRNA in situ hybridization or reporter gene expression, such as *CryBG (KH.S605.3)* and *Atp2a (KH.L116.40)*. Many ACC markers were conspicuously absent, but this may be due to the relatively early time point (12 hpf, late tailbud stage), well before hatching and ACC differentiation. This was a deliberate choice, as we were focused on papilla morphogenesis, which begins around this stage [22]. One resulting candidate *Islet* target revealed by RNAseq was *Astl-related (KH.C9.850)*, and its expression in the Islet+ cells of the papillae was confirmed by in situ hybridization (S8A Fig). Indeed, *Islet* knockout by CRISPR eliminated *Astl-related* reporter expression, supporting our approach to identifying new targets of Islet (S8B Fig). Furthermore, the top up-regulated gene by Islet overexpression (and 17th most down-regulated by *Islet* CRISPR) was *KY21.Chr10.318*, which encodes a Fibrillin-related (Fbn) protein. This gene was previously shown to be specifically expressed in the central *Islet+* cells by in situ hybridization [23], further validating our approach to identifying putative Islet target genes.

One particularly interesting ACC-specific candidate that was among the genes most highly up-regulated by Islet overexpression was *Villin (KH.C9.512)*, an ortholog of the *Villin* family of genes encoding effectors of actin regulators [55]. The apical extensions of the ACCs are highly enriched for actin filaments and microtubules [9], suggesting that cytoskeletal modulation may be important for the extended length of these cells relative to surrounding cells. We confirmed the expression of *Villin* in the papillae by in situ hybridization and reporter plasmids (Fig 7C and 7D). In the *Islet* CRISPR condition, *Villin* was the top down-regulated gene by *Islet* CRISPR knockout as well. *Villin* reporter expression was reduced in intensity but not completely lost upon knockout of *Islet* by CRISPR (Fig 7E and 7F), yet was dramatically up-regulated by Islet overexpression (Fig 7G and 7H). This suggests partially redundant activation of *Villin* by another factor, likely at earlier developmental stages (e.g., by *Foxc* or *Foxg*), and that Islet might be required for its sustained expression specifically in the central cells of the papilla throughout morphogenesis. This is consistent with the weak but broad expression of *Villin>GFP* in the entire papilla territory (Fig 7D), and the fact that papilla territory cells are already more elongated than epidermal cells in other parts of the embryo even at earlier stages [22].

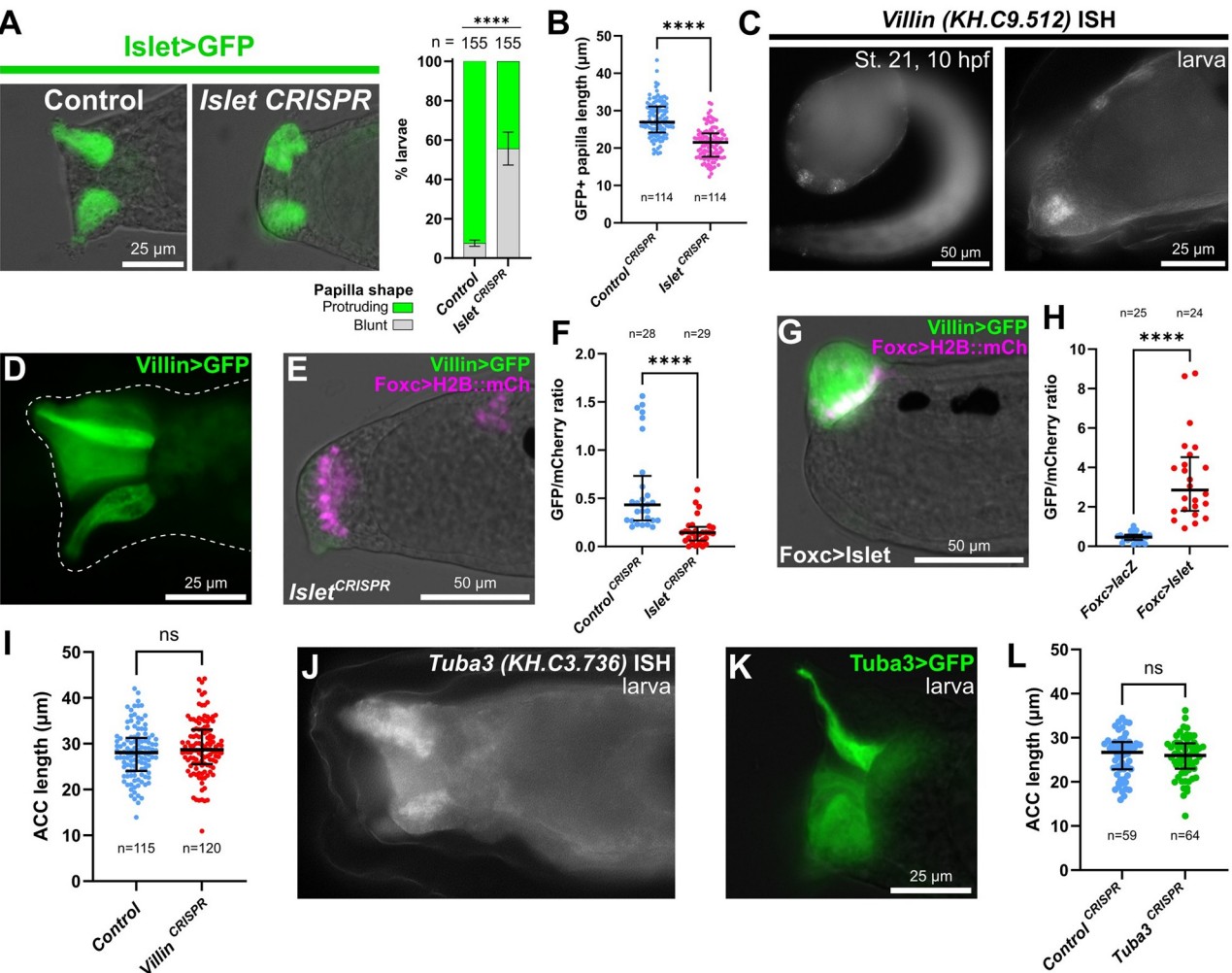

**Fig 7. Islet is also required for papilla morphogenesis.** (A) Papilla shape is shortened and blunt at the apical end upon tissue-specific CRISPR/Cas9-mediated mutagenesis of *Islet*. Embryos were electroporated with *Islet intron 1 + -473/-9>Unc-76::GFP* and *Foxc>Cas9*. *Islet* CRISPR was performed using *U6>Islet.2* sgRNA plasmid and the negative control used *U6>Control*. Larvae were imaged at 20 hpf/20 ˚C (~st. 28). Right: Scoring of percentage of GFP+ larvae classified as having normal "protruding" or blunt papillae, as represented to the left. Experiment was performed and scored in duplicate, using 2 different GFP fusions: Unc-76::GFP and DcxΔC::GFP [88]. Replicate 1: *n* = 100 for either condition; replicate 2: *n* = 55 for either condition **** *p* < 0.0001 in both duplicates as determined by Fisher's exact test. (B) Quantification of papilla cell (*Islet intron 1 +-473/-9>Unc-76::GFP* +) lengths along apical-basal axis in negative control and *Islet* CRISPR larvae at 18 hpf/20 ˚C (~st. 28). Both *Islet.1* and *Islet.2* sgRNAs used in combination. Statistical significance tested by unpaired *t* test (two-tailed). See S8 Fig for duplicate experiment. (C) In situ mRNA hybridization of *Villin*, showing expression in *Foxg+/Islet+* central papilla cells at 10 hpf/20 ˚C (st. 21, left) and at larval stage (~st. 27, right). (D) *Villin -1978/-1>Unc-76::GFP* showing expression in the papilla territory of electroporated larvae (~st. 28), strongest in the central cells. (E) *Villin -1978/-1>Unc-76::GFP* in st. 28 larvae is down-regulated by tissue-specific CRISPR/Cas9 mutagenesis of *Islet* (*Foxc>Cas9 + U6>Islet.1 + U6>Islet.2*, see text for details). (F) Quantification of effect of *Islet* CRISPR (as in panel E) on *Villin -1978/-1>Unc 76::GFP/Foxc>H2B::mCherry* mean fluorescence intensity ratios in ROIs defined by the mCherry+ nuclei (see Methods for details). Significance determined by Mann–Whitney test (two-tailed). (G) *Villin* reporter is up-regulated in st. 28 larvae by overexpressing Islet (*Foxc>Islet*, see text for details). (H) GFP/mCherry ratio quantification done in identical manner as in F, but comparing *Islet* overexpression (as in panel G) and control *lacZ* larvae. (I) Quantification of ACC lengths measured in negative control and papilla-specific *Villin* CRISPR larvae at 17 hpf/20 ˚C (~st. 27). Significance tested by unpaired *t* test (two-tailed). Although no statistically significant difference between control and CRISPR larvae was observed in this replicate, average ACC length was significantly shorter in the CRISPR condition in an additional replicate (S8 Fig). (J) mRNA in situ hybridization for *Tuba3*, showing enrichment in the central cells of the papillae in st. 27 larvae. (K) *Tuba3>Unc-76::GFP* reporter plasmid is broadly expressed in the papillae of st. 27 larvae but stronger in central cells. (L) Papilla-specific CRISPR knockout of *Tuba3* does not result in decrease of average ACC apical-basal cell length compared to negative control CRISPR using *U6>Control* sgRNA instead. Significance tested by unpaired *t* test (two-tailed). ns = not significant. All large bars indicate medians and smaller bars indicate interquartile ranges. See S3 and S4 Data for the data underlying the graphs and for statistical test details. ACC, axial columnar cell; ROI, region of interest; sgRNA, single-chain guide RNA.

To test whether *Villin* is required for proper morphogenesis of *Islet+* cells in the papilla, we performed tissue-specific CRISPR knockout using a combination of 3 validated sgRNAs spanning most of the coding sequence (S3E Fig). Because the functionally important "headpiece" domain is encoded by the last exon, we combined an sgRNA targeting this exon with 2 sgRNAs targeting more upstream exons. In one batch of *Villin* CRISPR larvae, ACCs were not significantly shorter on average along the apical-basal axis than in control larvae (Fig 7I). However, average ACC length was significantly shorter in CRISPR larvae than control larvae in a duplicate experiment (S8E Fig). This contrast between replicates was found to be entirely due to variability between batches of control larvae, not the *Villin* CRISPR larvae (S8F Fig). Because the ACCs have been shown to dynamically extend or contract in length, possibly in response to external stimuli [56,57], we suspect that these differences in average length in different batches of control animals are due to as of yet unidentified environmental conditions.

In addition to actin regulators, we searched our list of putative Islet targets for microtubule components and regulators, since microtubule bundles were reported in the apical protrusions of the ACCs in *Distaplia occidentalis* [58]. We identified a gene encoding a divergent Tubulin alpha monomer (*Tuba3*, *KH.C3.736*) as one such potential target. Enrichment of *Tuba3* expression in the central papillae was confirmed by in situ hybridization (Fig 7J) and a Tuba3 reporter plasmid (Fig 7K). However, papilla-specific *CRISPR* knockout of *Tuba3* did not result in significantly shorter ACCs either (Fig 7L). Taken together, these results suggest that Islet is required for proper papilla morphogenesis, and that this may be due to its role in activating the expression of numerous effector genes. However, knocking out individual candidate effector genes like *Villin* or *Tuba3* has not yet revealed a key role for any one of these putative downstream targets.

## An investigation into the cell and molecular basis of larval settlement and metamorphosis

With our different CRISPR knockouts affecting different cell types of the papillae, we asked how these different perturbations might affect larval metamorphosis. Only the involvement of the PNs in triggering metamorphosis has been demonstrated [16,17], but it is not yet known how the regulatory networks and cell types of the papillae affect different processes during metamorphosis. We performed papilla-specific CRISPR as above using the *Foxc>Cas9* vector, targeting the 4 different transcription factors we have shown to be involved in patterning the cell types of the papillae: *Pou4*, *Islet*, *Foxg*, and *Sp6/7/8*. We assayed tail retraction and body rotation at the last stage of metamorphosis [28] (Fig 8A and 8B), as these are 2 processes that can be uncoupled in certain genetic perturbations or naturally occurring mutants [59].

Knockout of *Pou4* recapitulated recent published results on this transcription factor [16]. Namely, both tail retraction and body rotation were blocked in the vast majority of individuals. This suggests that proper specification and/or differentiation of PNs by *Pou4* is crucial for the ability of the larva to trigger the onset of metamorphosis. In contrast, *Islet* knockout did not affect tail retraction, but body rotation appeared somewhat impaired. This suggested that ACCs/ICs are not required for tail retraction, but might play a role in regulating body rotation downstream of it. Eliminating ACCs using *Islet>Sp6/7/8* had no effect on either tail retraction or body rotation (Fig 8C), confirming that ACCs are not required for metamorphosis, but that perhaps certain Islet targets might specifically regulate body rotation. Unsurprisingly, *Foxg* knockout modestly impaired both tail retraction and body rotation (Fig 8B and 8D, and S9 Fig), but also resulted in a noticeable fraction (approximately 19% on average) of "tailed juveniles" in which body rotation begins even in the absence of tail retraction. This unusual effect was seen even when repeating the experiment independently a third time, revealing consistent

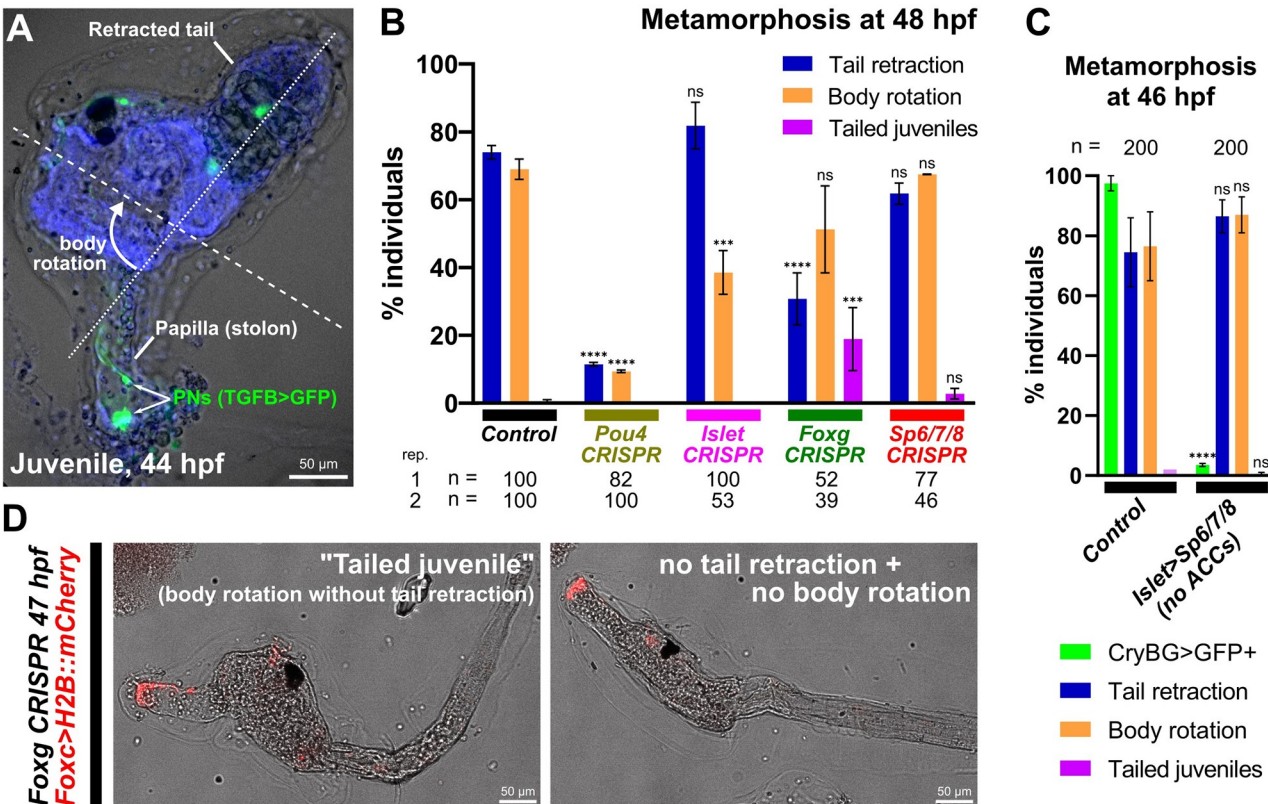

**Fig 8. Genetic perturbations of metamorphosis.** (A) *Ciona robusta* juvenile undergoing metamorphosis, showing the retracted tail and rotated anterior-posterior body axis (dashed lines). PNs in the former papilla (now substrate attachment stolon, or holdfast) labeled by *TGFB>Unc-76::GFP* (green). Animal counterstained with DAPI (blue). (B) Scoring of Foxc>H2B::mCherry+ individuals showing tail retraction and/or body rotation at 48 hpf/20 ˚C in various papilla territory-specific (using *Foxc>Cas9*) CRISPR-based gene knockouts. Experiments were performed and scored in duplicate and percentages averaged, except for *Foxg* CRISPR for which a third replicate was performed (see S9 Fig). Number scored individuals in each replicate indicated underneath. "Tailed juveniles" have undergone body rotation but not tail retraction, whereas normally body rotation follows tail retraction. The sgRNA plasmids used for each condition were as follows- Control: *U6>Control; Pou4: U6>Pou4.3.21 + U6>Pou4.4.106; Islet: U6>Islet.2; Foxg: U6>Foxg.1.116 + U6>Foxg.5.419; Sp6/7/8: U6>Sp6/7/8.4.29 + U6>Sp6/7/8.8.117*. (C) Plot showing lack of any discernable metamorphosis defect after eliminating ACCs using *Islet intron 1 + bpFOG>Sp6/7/8* (images not shown). Only *Islet intron 1 + bpFOG>H2B::mCherry+* individuals were scored. Experiment was performed and scored in duplicate and averaged (*n* = 100 each duplicate). ACC specification was scored using the *CryBG>Unc-76::GFP* reporter. (D) Example of "tailed juveniles" at 47 hpf/20 ˚C compared to a larva in which no tail retraction or body rotation has occurred, elicited by tissue-specific *Foxg* CRISPR (*Foxc>Cas9 + U6>Foxg.1.116 + U6>Foxg5.419*). See S9 Fig for scoring. All error bars denote upper and lower limits. **** *p* < 0.0001, *** *p* < 0.0015 in both duplicates, ns = not significant in at least 1 duplicate, as determined by chi-square test comparing to the control conditions. See S4 Data for the data underlying the graphs and for statistical test details. ACC, axial columnar cell; PN, papilla neuron; sgRNA, single-chain guide RNA.

uncoupling of these 2 processes upon *Foxg* knockout (S9 Fig). Finally, *Sp6/7/8* CRISPR did not substantially alter either tail retraction or body rotation. Taken together, these results paint a more complex picture of regulation of metamorphosis by the papillae. Our findings suggest that different cell types of the papillae might play distinct roles in the regulation of metamorphosis, perhaps interacting with one another to regulate different steps, or that certain transcription factors might be required for the expression of key rate-limiting components of these different processes. Further work will be required to disentangle these different cellular and genetic factors, which we hope will be aided by our cell type-specific reporters and CRISPR reagents.

## Discussion

Sensory systems are crucial for interactions between organisms and their environment. The concentration of sensory functions in the head is thought to have played a central role in vertebrate evolution, leading to a more active behavior emerging from early filter-feeding chordate ancestors [60–62]. The peripheral components of the sensory systems in vertebrates arise from 2 physically close but distinct ectodermal cell populations, the cranial sensory placodes and the neural crest [63]. Cranial sensory placodes are characterized by their common ontogenetic origin from a crescent-shaped region surrounding the anterior neural plate. Our understanding of the evolutionary origins of structures long presented as vertebrate novelties has benefited from an increasing number of comparative studies with tunicates. Several discrete populations of peripheral sensory cells originating from distinct ectodermal regions in tunicates have respectively been linked to neural crest and cranial placodes, among them the sensory adhesive papillae [9,64–67].

Our results have confirmed the existence of molecularly distinct cell types in the *Ciona* papillae and the developmental pathways that specify them (summarized in Fig 9). Using CRISPR/Cas9-mediated mutagenesis, we have shown that different transcription factors are required for their specification, differentiation, and morphogenesis. Namely, ACCs and ICs are specified from Foxg+/Islet+ cells at the center of each of the 3 papillae, while OCs and PNs are specified from interleaved Islet-negative cells that nonetheless derive from initially Foxg + cells. While Sp6/7/8 specifies IC versus ACC fate among Islet+ cells, Delta/Notch signaling suppresses PN fate and promotes OC fate among Islet-negative cells. While there appear to be 2 molecularly distinct collocyte subtypes (OCs and ICs), both contain granules that are stained by PNA, and therefore both are likely to be involved in glue production. Where they differ might be in the timing of glue production and/or secretion, as they showed distinct subcellular localization of PNA+ granules, and PNA production was previously shown to start very early [9].

Our results also demonstrate a clear distinction between CryBG+ ACCs and Pou4+ PNs. Previously, these cells types have been confused and only recently distinguished by TEM and different molecular markers [9]. Here we show that, while both arise from Foxc+/Foxg+ cells, ACCs are not specified by Pou4, and PNs are not specified by *Islet*. However, because *Pou4* can activate *Foxg* expression in a proposed feedback loop [68], overexpression of *Pou4* might result in ectopic activation of ACC markers via ectopic *Foxg* and *Islet* activation.

There are still unanswered questions that we hope future work will address:

(1) *How do the 3 "spots" of Foxg+/Islet+ cells form in an invariant manner*? Ephrin-Eph signaling is often responsible for suppression of FGF/MAPK signaling in alternating cells in *Ciona* embryos, via asymmetric inheritance/activation of p120 RasGAP [69,70]. This is also true in the earlier patterning of the papilla territory, where EphrinA.d suppresses FGF/MAPK to promote *Foxg* activation [21]. Curiously, later expression of *EphrinA.d* in the lineage appears to be stronger in medial *Foxg*+ cells than in lateral cells [21]. This distribution would suffice to result in the alternating ON/OFF pattern of MAPK activation at the tailbud stage that results in the 3 foci of *Foxg/Islet* expression. Thus, it may be informative to test the ongoing functions of Ephrin-Eph signaling in this lineage throughout development. Interestingly, we did not observe substantial loss of PNs in *Sp6/7/8* CRISPR larvae, even though Sp6/7/8 down-regulates *Foxg* in between the 3 "spots" [21] and is necessary for OC reporter expression (Fig 4F). It is possible that FGF/MAPK suppresses *Islet* in parallel, and that loss of *Sp6/7/8* is not sufficient to expand *Islet* expression and ACC/IC progenitor fate (at the expense of PN/OC progenitors) in the same manner that the MEK inhibitor U0126 does.

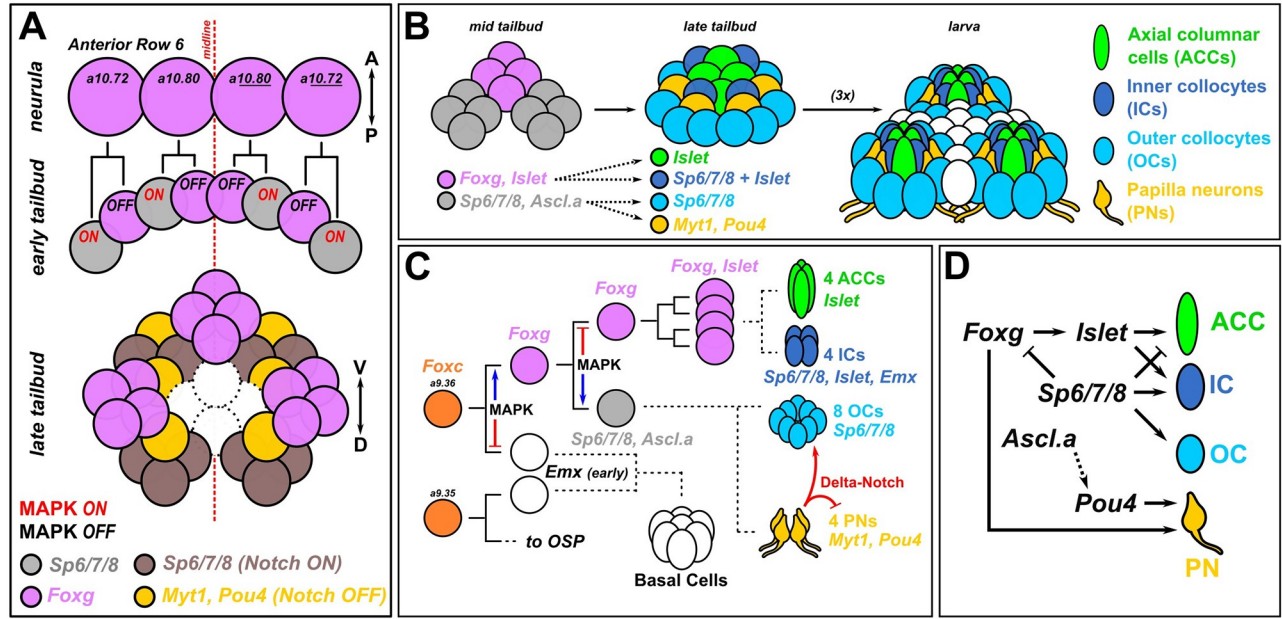

**Fig 9. Summary diagram.** (A) Updated diagram of the development of the anterior descendants of Row 6 in the neural plate to show the proposed patterns of MAPK and Delta/Notch signaling that set up the 3 *Foxg*+ clusters and interleaved *Foxg*-negative neurogenic cells. (B) Diagram proposing the contributions of *Foxg*+ and *Foxg*-negative cells to later patterns of transcription factors that specify the different cell types found in each papilla, which is in turn is repeated 3 times, thanks to the process shown in panel A. (C) Papilla development shown as cell lineages, with dashed lines indicating uncertain cell divisions and lineage history. MAPK regulates binary fate choices, promoting *Foxg* expression in the papillae proper at first but later suppressing it (and *Islet*). Lastly, Delta-Notch signaling promotes OC fate and limits PN specification through lateral inhibition. Cell type numbers based on [9]. (D) Provisional gene regulatory network diagram of the transcription factors involved in specification and differentiation of the different papilla cell types. Arrowheads indicate activating gene expression or promoting cell fate, while blunt ends indicate repression of gene expression of cell fate. Solid lines indicate regulatory links (direct or indirect) that are supported by the current data and literature. Dashed lines indicate regulatory links that have not been tested, or need to be investigated in more detail. A-P, anterior-posterior; D-V, dorsal-ventral; OC, outer collocyte; OSP, oral siphon primordium; PN, papilla neuron.

(2) *How are PNs specified adjacent to the Islet+ cells*? Since Delta/Notch signaling is involved in PN versus OC fate, we propose that there is something that biases Notch signaling to be activated preferentially in those cells not touching the *Islet*+ cells. This could be due to cell-autonomous activation of Notch signaling in the *Islet*+ cells, which in turn would allow for suppression of Notch in adjacent cells fated to become PNs. A recent study showed *Pou4* expansion with concomitant "U"-shaped expansion of *Islet* in larvae with a single, enlarged papilla when inhibiting BMP signaling [23]. However, our expansion of *Islet* with treatment of U0126 (based on experiments from Wagner and colleagues) suggests the opposite, the elimination of *Pou4*+ PNs. Why the discrepancy? One possibility is that inhibiting BMP results in specification of supernumerary RTEN-like neurons from adjacent epidermis, not PNs. While *Pou4* is expressed in all epidermal neurons including PNs and RTENs, our pre-ferred PN marker *KH.C4.78* is not expressed in RTENs. However, we did not observe either loss or expansion of *C4.78>GFP* in larvae with enlarged, single papillae resulting from treatment with the BMP inhibitor DMH1 (S10 Fig). This suggests that inhibition of FGF/MEK and BMP have slightly different effects on patterning and neurogenesis in the papillae. Further work will be needed to resolve these and other intriguing nuances.

(3) *What activates the expression of Sp6/7/8 in Islet+ cells, ultimately promoting IC specification*? We do not yet know the exact mitotic history of the ACCs/ICs. How do the initially four Islet+ cells divide, and which daughter cells give rise to ACCs versus ICs? Are ACCs/ICs

specified in an invariant manner, or is there some variability? Finally, what allows the "creeping" activation of *Sp6/7/8* in the outer ring of cells that likely become the ICs? Is this due to additional asymmetric FGF/MAPK activation downstream of Ephrin-Eph? Or could this be due to some other signaling pathway? Is there an inductive signal from adjacent cells, for instance PNs or common PN/OC progenitors?

(4) *How do the different papilla cell types regulate metamorphosis*? We noticed some uncoupling of tail resorption and body rotation upon targeting different transcription factors for deletion in the papillae (Fig 8). This was most apparent in the *Foxg* knockout, in which a substantial portion of individuals displayed the "tailed juvenile" phenotype in which body rotation proceeds even in the absence of tail resorption. From the *Pou4* knockout, it is clear that PNs are upstream of both tail resorption and body rotation, but the partial uncoupling seen with the other manipulations were particularly intriguing. This uncoupling has been reported before in *Cellulose synthase* mutants, which results in similar tailed juveniles [71]. Additionally, perturbation of Gonadotropin-releasing hormone (GnRH) or the prohormone convertase enzyme (PC2) necessary for its processing similarly blocks tail resorption but not body rotation and further adult organ growth [72]. Thus, it is possible that while Pou4 disrupts PN specification altogether, Foxg might be more specifically required for GnRH or other neuropeptide expression/processing in the PNs. Supporting this idea, the *Foxg* CRISPR did not disrupt PN specification (as assayed by *TGFB>GFP* reporter expression) as robustly as did *Pou4* CRISPR. Alternatively, this uncoupling may also be a result of the different roles of Foxg in regulating different papilla cell types, which may be unequally and variably affected due to CRISPR knockout mosaicism in F0. Finally, the appearance of juveniles with resorbed tails but no further body rotation in the *Islet* CRISPR condition suggests a crucial role for the "core" cells of the papilla (ACCs/ICs) in metamorphosis downstream of PNs. However, body rotation was not affected by eliminating either ICs (*Sp6/7/8* CRISPR, Fig 8B) or ACCs (*Islet>Sp6/7/8*, Fig 8C), suggesting Islet is required for the expression of a "body rotation" factor independently of IC/ACC specification. Clearly, more work will be required to understand the contributions of different cell types, and potentially different molecular pathways in the same cell type, towards either activation or suppression of specific body plan rearrangement processes in tunicate larval metamorphosis.

(5) *Are the tunicate larval papillae homologous to vertebrate cement glands and/or olfactory placodes*? The papillae have often been compared to the cement glands of fish and amphibian larvae, which are transient adhesive organs secreting sticky mucus [73]. Even though they are innervated by trigeminal fibers, the secreting cells from the cement gland differentiate from a surface ectoderm region anterior to the oral ectoderm and the panplacodal domain [74]. Therefore, they are usually not considered placodal derivatives. Despite their variability in size, number, and location, head adhesive organs are proposed to be homologous across vertebrate species based on their shared expression of Pitx1/2 and BMP4 genes, innervation by trigeminal fibers, and inhibiting mechanism of swimming behavior [73,74]. While recent papers have revealed an important role for BMP signaling in pattering the tunicate papilla territory [23,24], suggesting a potential evolutionary connection, additional work on the molecular basis of the papillary glue in tunicates will be required to answer questions of homology between these adhesive organs. Our identification of molecular signatures for both collocyte subtypes in the papillae of *Ciona* provides a starting point for future investigations and may allow for broader evolutionary comparisons between chordates and other bilaterians. If one considers the vertebrate cement gland and the trigeminal

neurons that innervate it as a functional unit but no homology to the tunicate papillae can be established, they might represent an interesting case of evolutionary convergence.

Besides adhesion, a key role of the papilla is to act as an organ for bimodal mechano- and chemo-sensation regulating larval settlement and metamorphosis [75]. It has been proposed that the papilla territory may be a homolog of the olfactory placode, based on the expression of several regulators investigated in this study including *Sp6/7/8*, *Islet*, *Foxc*, and *Foxg* [18,21]. The roles of Foxg in specifying the papilla territory may also be linked evolutionarily to the functions of Foxg1 in vertebrate olfactory development [76,77]. Intriguingly, the specification of sensory PNs and probable adhesive-secreting OCs from shared *Ascl.a+* progenitors (as revealed by Notch pathway inhibition experiments, Fig 6) suggests a close regulatory link between sensory/adhesive functions in *Ciona* papillae. In vertebrates, Ascl1 not only regulates sensory cell specification in developing olfactory epithelium and taste buds [78–80] but is also required for intestinal secretory cells [81,82]. Additionally, *Ascl3* is expressed in vertebrate salivary gland duct progenitors, which are also highly enriched for orthologs of other tunicate papilla regulators like *Foxc*, *Sp6/7/8*, and *Islet* [83,84]. Thus, while the papillae of *Ciona* might represent a tunicate-specific evolutionary novelty, overlaying unrelated networks for sensory neurons and adhesive-secreting cells together in a single embryonic territory, it is also possible that they rely on a shared sensory/exocrine program that might have deeper evolutionary origins instead [85].

## Supporting information

**S1 Fig. Finding papilla cell type-specific markers in single-cell RNAseq data.** (A) Cell clusters based from reanalysis and re-clustering of whole-larva single-cell RNA sequencing (scRNAseq) data from Cao and colleagues (see S1 Data). Dashed red box indicated clusters 3 and 33, which appeared to correspond to several papilla cell types. (B) Cells from clusters 3 and 33 from plot A set aside and re-clustered. (C) Differential expression plots showing examples of candidate papilla cell type marker genes mapped onto clusters in B. (D) Fluorescent, whole-mount in situ mRNA hybridization (green) for certain genes plotted in C, labeling different cells in the papillae of *Ciona robusta* (*intestinalis* Type A) hatched larvae. (E) Protein domain prediction diagrams for select cell type-specific marker proteins generated by SMART [89]. Unless specifically named, genes are indicated by KyotoHoya (KH) ID numbers (e.g., *KH.L96.43*). All larvae were fixed at 18 h post-fertilization (hpf), 20 ˚C, except for *C11.360* and *C2.1013* (18.5 hpf). Blue counterstain is DAPI.
(TIF)

**S2 Fig. Additional marker genes and reporter plasmids expressed in papillae.** (A) *TGFB>Unc-76*::*GFP* reporter (green) is not co-expressed in the same cells as the *Islet intron 1 + -473/-9>mCherry* reporter (pink) at 20.5 hpf (~st. 29). (B) Cross-talk between *CryBG>Unc-76*::*GFP* and *TGFB>Unc-76*::*mCherry* reporter plasmids at 16 hpf (~st. 26), showing aberrant co-expression in ACCs and/or PNs only when co-electroporated. (C) Mutually exclusive expression of *CryBG>lacZ* in ACCs (cyan), *Gnrh1>Unc-76*::*GFP* in PNs (yellow), and *C11.360>Unc-76*::*mCherry* in ICs (magenta), with DAPI counterstained in gray. This larva is the same as in main Fig 2G, with an additional channel and different false coloring. (D) Images from Fig 2 with mCherry and GFP channels displayed separately. (E) *C. intestinalis* (Type B) larva electroporated with *C. robusta C11.360>Unc-76*::*GFP* reporter plasmid, showing specific but weak expression. (F) *C. intestinalis* (Type B) larva electroporated with *C. robusta L96.43>Unc-76*::*GFP* reporter plasmid, also showing weak expression. Papillae in panels E and F outlined by dashed lines. (G) Reporter plasmid containing the first intronic region of *Emx*

drives expression in ICs at 20 hpf (~st. 29), likely corresponding to the "ring" of late *Emx* expression in *Islet+* cells reported in Wagner and colleagues and distinct from earlier *Emx* expression in the papilla lineage as described in Liu and Satou. (H) *C14.116>Unc-76::mCherry* reporter expressed in central cells (ACCs+ICs, pink) and basal cells around the 3 papillae at 20.5 hpf (~st. 29). (I) Immunostaining for the Flag epitope tag fused to the Islet-rescue protein used for *Islet>Islet* experiments in Fig 4. Flag immunostaining in green and *Foxc>H2B:: mCherry* in pink in merged image. Larvae fixed at 19 hpf (~st. 28). DAPI in gray. ACCs, axial columnar cells; PN, papilla neuron.
(TIF)

**S3 Fig. Validation of sgRNAs for CRISPR/Cas9-mediated mutagenesis.** Gene loci diagrams for the 4 transcription factor-encoding genes investigated in this study: *Sp6/7/8*, *Foxg*, *Islet*, and *Pou4*. Plots underneath each gene show validation by Illumina sequencing ("Next-generation sequencing" or NGS) of amplicons, performed as "Amplicon-EZ" service by Azenta. Mutagenesis efficacies are calculated by this service, and histograms of mapped reads show specificity of indels elicited by each sgRNA. Negative control amplicons are amplified from samples that were electroporated with no sgRNA, *U6>Control* sgRNA, or sgRNAs targeting unrelated amplicon regions. Note different y axis scales for each plot. Asterisks in *Villin* exon 5 and *Tuba3* amplicon plots indicate naturally occurring indels. Precise calculation of mutagenesis efficacy for *Villin.5.105* and *Tuba3.3.24* sgRNAs was not given due to these natural indels.
(TIF)

**S4 Fig. Effect of various CRISPR knockouts on specification of ACCs, PNs, and OCs.** (A) Scoring of effect of papilla-specific CRISPR knockout of *Foxg* or *Pou4* on specification of ACCs and PNs. Embryos were electroporated with *Foxc>H2B::mCherry*, *Foxc>Cas9*, *CryBG>Unc-76::GFP* (ACC reporter), *TGFB>Unc-76::GFP* (PN reporter), or *L141.36>Unc-76::GFP* (OC reporter), and gene-specific sgRNA combinations (see below for specific combinations). All were performed in duplicate and scores averaged, but some replicates and conditions are represented in Figs 3 and 4 also. Total embryos ranged between 76 and 100 per condition per replicate. Specific sgRNAs used: *Foxg*: *U6>Foxg.1.116 + U6>Foxg.5.419*; *Pou4*: *U6>Pou4.3.21 + U6>Pou4.4.106*; *Sp6/7/8*: *U6>Sp6/7/8.4.29 + U6>Sp6/7/8.8.117*; *Islet*: *U6>Islet.2*; *Control*: *U6>Control*. (B) *Foxg*, *Pou4*, and *Sp6/7/8* sgRNAs were also tested alone (as opposed to pairs in combination) using reporter assays as in Figs 3 and 4. Those sgRNAs used further are highlighted in blue font. Additional sgRNAs abandoned due to low efficacy indicated in black font. For all plots, only larvae showing *Foxc>H2B::mCherry* expression in the papillae were scored. See S4 Data for the data underlying the graphs and for statistical test details.
(TIF)

**S5 Fig. CRISPR rescues and possible expansion of ICs in *Islet>Sp6/7/8*.** (A) *CryBG>GFP* expression in *Islet* (left) and *Foxg* (right) CRISPR larvae is rescued by co-electroporation with *Islet intron 1 + bpFOG>Flag::Islet-rescue* or *Foxg>Foxg-rescue* constructs, respectively, thanks to silent point mutations disrupting the sgRNA target binding sites. (B) Expression of *C11.360>GFP* is rescued in *Sp6/7/8* CRISPR larvae upon co-electroporation with an *Islet intron 1 + bpFOG>Sp6/7/8-rescue* construct. (C) Example of expanded IC reporter (*C11.360>Unc-76::GFP*, green) in larvae (20 hpf/20 °C, ~st. 29) electroporated with *Islet intron 1 + -473/-9>Sp6/7/8*, as determined by perfect overlap with the *Islet intron 1 + -473/-9>H2B:: mCherry* reporter (pink). See text for more details. See S1 File for exact sequences and detailed electroporation recipes. See S4 Data for the data underlying the graphs and for statistical test

details.
(TIF)

**S6 Fig. Single-channel images of PNA-stained larvae shown in Fig 5.**
(TIF)

**S7 Fig. Delta/Notch components and lack of SUH-DBM effect on ACC/IC fate choice.** (A) Fluorescent whole-mount in situ mRNA hybridization for *Delta like non-canonical Notch ligand (KH.L50.6)* in a stage 23 embryo, marking epidermal sensory neurons including papilla neurons (arrows) and caudal epidermal neurons (CENs). (B) No significant difference in expression of either *CryBG>mCherry* or *C11.360>GFP* in larvae at 19 hpf at 20 ˚C (~st. 28) electroporated with the Delta-Notch pathway-inhibiting *Foxc>SUH-DBM*. Right: representative panels showing expression of both reporters in control and SUH-DBM-expressing larvae. See S4 Data for the data underlying the graphs and statistical test details.
(TIF)

**S8 Fig. Investigating the regulation of papilla morphogenesis by Islet and its putative transcriptional targets.** (A) In situ mRNA hybridization (ISH) showing expression of *Astl-related* (green) specifically in the *Islet+* cells of the papillae (labeled by *Islet intron 1 + -473/-9>mCherry*, pink nuclei). (B) Tissue-specific CRISPR/Cas-mediated mutagenesis of *Islet* results in loss of *Astl-related>Unc-76::GFP* reporter expression in ACCs/ICs (green). *Foxc>Cas9* was used to restrict CRISPR/Cas9 to the papilla territory (labeled by *Foxc>H2B::mCherry*, pink nuclei). Asterisk denotes residual reporter expression in cells outside the papilla territory. Right: Scoring of larvae represented in panel B, following criteria used for Fig 3. **** $p < 0.0001$ using chi-square test. (C) Second duplicate of *Islet* CRISPR experiment in Fig 7B. Statistical significance was determined by unpaired $t$ test (two-tailed). (D) Representative images of control and *Islet* CRISPR larvae used for measurements in Fig 7B, with example of apical-basal cell length measurements. (E) Both duplicates of *Villin* CRISPR experiments side by side. Statistical significance was calculated using Mann–Whitney test (two-tailed) for replicate 1 and unpaired $t$ test (two-tailed) for replicate 2. ns = not significant. (F) Comparison of ACC lengths measured in control larvae from the 2 duplicate *Villin* CRISPR experiments, showing statistically significant difference in average ACC lengths between different batches of larvae, calculated using Mann–Whitney test (two-tailed). See S3 and S4 Data for the data underlying the graphs and for statistical test details.
(TIF)

**S9 Fig. Third replicate of *Foxg* CRISPR effects on metamorphosis.** Scoring of Foxc>H2B::mCherry+ individuals as represented in Fig 7D, in third replicate of data in Fig 7B. See S1 File for detailed plasmid electroporation recipes. **** $p < 0.0001$ calculated by Fisher's exact test. See S4 Data for the data underlying the graphs and for statistical test details.
(TIF)

**S10 Fig. Pharmacological inhibition of BMP signaling and papilla neuron specification.** Larva co-electroporated with *Islet intron 1 + -473/-9>mCherry* (pink) and PN reporter *C4.78>Unc-76::GFP* (green) showing lack of substantial loss or expansion of PNs in larvae treated with the BMP pathway inhibitor DMH1, in spite of expanded *Islet* reporter expression and a single, enlarged papilla. Left: scoring of PN reporter expression in Islet>mCherry + DMSO (negative control) and DMH1-treated larvae. All larvae raised to 19 hpf at 20 ˚C (~st. 28). See S4 Data for the data underlying the graphs and for statistical test details.
(TIF)

**S1 Data. Differential gene expression table of re-analyzed whole larva single-cell RNAseq data from Cao and colleagues (new clusters 3 and 33 subclustered into 11 subclusters).**
(XLSX)

**S2 Data. Differential gene expression table of bulk RNAseq analysis of *Islet* overexpression or *Islet* CRISPR embryos vs. control embryos.**
(XLSX)

**S3 Data. Papilla length measurements in different CRISPR and control larvae.**
(XLSX)

**S4 Data. Summary of statistical tests of proportions (i.e., scoring).**
(XLSX)

**S1 Movie. Confocal stack represented by Fig 5A.**
(AVI)

**S2 Movie. Confocal stack represented by Fig 5B.**
(AVI)

**S1 File. All relevant DNA and protein sequences used in this study (supplemental sequence file).**
(DOCX)

## Acknowledgments

We thank members of the labs at Georgia Tech and Innsbruck for critical feedback and support. We thank Susanne Gibboney, Tanner Shearer, Alex Gurgis, Lindsey Cohen, Akhil Kulkarni, and Eduardo Gigante for technical assistance.

## Author Contributions

**Conceptualization:** Christopher J. Johnson, Florian Razy-Krajka, Fan Zeng, Ute Rothbächer, Alberto Stolfi.

**Data curation:** Christopher J. Johnson, Florian Razy-Krajka, Fan Zeng, Katarzyna M. Piekarz, Ute Rothbächer, Alberto Stolfi.

**Formal analysis:** Christopher J. Johnson, Florian Razy-Krajka, Fan Zeng, Katarzyna M. Piekarz, Ute Rothbächer, Alberto Stolfi.

**Funding acquisition:** Christopher J. Johnson, Ute Rothbächer, Alberto Stolfi.

**Investigation:** Christopher J. Johnson, Florian Razy-Krajka, Fan Zeng, Ute Rothbächer, Alberto Stolfi.

**Methodology:** Christopher J. Johnson, Florian Razy-Krajka, Fan Zeng, Katarzyna M. Piekarz, Shweta Biliya, Ute Rothbächer, Alberto Stolfi.

**Project administration:** Christopher J. Johnson, Florian Razy-Krajka, Fan Zeng, Ute Rothbächer, Alberto Stolfi.

**Software:** Katarzyna M. Piekarz.

**Supervision:** Florian Razy-Krajka, Ute Rothbächer, Alberto Stolfi.

**Validation:** Christopher J. Johnson, Florian Razy-Krajka, Fan Zeng, Katarzyna M. Piekarz, Ute Rothbächer, Alberto Stolfi.

**Visualization:** Christopher J. Johnson, Florian Razy-Krajka, Fan Zeng, Katarzyna M. Piekarz, Ute Rothbächer, Alberto Stolfi.

**Writing – original draft:** Katarzyna M. Piekarz, Shweta Biliya, Ute Rothbächer, Alberto Stolfi.

**Writing – review & editing:** Christopher J. Johnson, Florian Razy-Krajka, Fan Zeng, Katarzyna M. Piekarz, Shweta Biliya, Ute Rothbächer, Alberto Stolfi.

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
