## [Editor Report · Decision Letter 0]

14 May 2023

Dear Dr Stolfi, 

Thank you for submitting your manuscript entitled "Specification of distinct cell types in a sensory-adhesive organ for metamorphosis in the Ciona larva" for consideration as a Research Article by PLOS Biology.

Your manuscript has now been evaluated by the PLOS Biology editorial staff as well as by an academic editor with relevant expertise and I am writing to let you know that we would like to send your submission out for external peer review.

Once your full submission is complete, your paper will undergo a series of checks in preparation for peer review. After your manuscript has passed the checks it will be sent out for review. To provide the metadata for your submission, please Login to Editorial Manager (https://www.editorialmanager.com/pbiology) within two working days, i.e. by May 16 2023 11:59PM.

Kind regards,

Ines

--

Ines Alvarez-Garcia, PhD

Senior Editor

PLOS Biology

---

## [Decision Letter · Decision Letter 1]

9 Jul 2023

Dear Dr Stolfi,

Thank you for your patience while your manuscript entitled "Specification of distinct cell types in a sensory-adhesive organ for metamorphosis in the Ciona larva" was peer-reviewed at PLOS Biology. Please also accept my apologies for the delay in providing you with our decision. The manuscript has been evaluated by the PLOS Biology editors, an Academic Editor with relevant expertise, and by two independent reviewers. 

The reviews are attached below. As you will see, the reviewers find the results interesting and significant for the field, but they also raise several issues that would need to be addressed before we consider the manuscript for publication. Reviewer 1 raises concerns regarding off-target effects of CRISPR/Cas9-mediated mutagenesis and would like to see validation of the phenotypes observed or rescue experiments. In addition, this reviewer asks for the potential evolutionary implications of the findings among other issues. Reviewer 2 makes some suggestions to strengthen the study and points out several issues that would need to be clarified.

In light of the reviews and discussion with the Academic Editor, we would like to invite you to revise the work to thoroughly address the reviewers' reports. Given the extent of revision needed, we cannot make a decision about publication until we have seen the revised manuscript and your response to the reviewers' comments. Your revised manuscript is likely to be sent for further evaluation by all or a subset of the reviewers.

**IMPORTANT - SUBMITTING YOUR REVISION**

3. Resubmission Checklist

a) *PLOS Data Policy*

b) *Published Peer Review*

Sincerely,

Ines

--

Ines Alvarez-Garcia, PhD

Senior Editor

PLOS Biology

Reviewers' comments

Rev. 1:

In this paper, Johnson et al. examine the developmental genetics of sensory/adhesive organs - papillae - of tunicate larvae using Ciona. First, they re-analyzed previously published single cell RNA-seq data, and identified new molecular markers for papilla cell types - namely, axial columnar cells (ACCs), inner collocytes (ICs), outer collocytes (OCs), and papilla neurons (PNs). Using these markers in a series of genetic perturbation experiments involving papilla-specific CRISPR/Cas9-mediated mutagenesis, they investigate how the papilla cell types are specified. They find, in particular, that Islet and Sp678 combinatorially specifiy the fates of ACCs, ICs and OCs, while Notch signaling inhibits PN differentiation and promotes OC differentiation. In addition, the authors report the effects of genetic perturbation of papilla development on metamorphosis. Integrating the results of their study and others, the authors reconstruct a provisional gene regulatory network underpinning papilla development in Ciona.

I think that this paper contributes nicely to filling the basic knowledge gap about the development of paplillae in Ciona - the mechanism of cell type differentiation. Experiments are appropriately designed. I also found that the paper was, for the most part, clearly written and had a logical structure. The paper should certainly be of interest to biologists working on Ciona development and neurobiology. I think that this paper could be of interest more broadly to evolutionary developmental biologists working on other systems. However, evolutionary implications of the findings are not discussed, and therefore I found the manuscript in its current form to be decisively Ciona-centric. I have listed my major and minor concerns below.

Major comments:

1) Off-target effects of CRISPR/Cas9-mediated mutagenesis are not adequately addressed. The authors should consider validating each knockout/knockdown phenotype, for instance, by using another set of sgRNAs designed to target the same gene but differ in sequence from the ones used, or by performing genetic rescue experiments.

2) Statistical significance of difference is not assessed in any of the experiments except for one (Fig. 7H). Thus, claims such as that expression was "substantially reduced relative to the control" (p.10 line 285) are not substantiated. In some cases, no quantification is provided at all. For example, the effects of U0126 on PN and OC marker expression, and expansion of the PN reporter expression by Foxc>SUH-DBM are not quantified or statistically assessed. Statistical assessment of significance is essential for evaluating the strength of support for each hypothesis being tested.

3) Evolutionary implications of the findings are unclear. The authors mention (in Discussion) hypotheses of homology between tunicate larval papillae and vertebrate structures (e.g. cement glands). However, it is unclear whether the findings of this study shed any insights relative to these hypotheses beyond providing resources or "a starting point for future investigations". Therefore, the findings of this study by themselves do not appear to have a broad impact beyond the field of ascidian development and neurobiology.

4) Mutually exclusive expression and co-expression of genes are often unclear based on images presented in the figures. I suggest showing individual channels separately whenever demonstrating mutually exclusive expression patterns or co-expression patterns (e.g. Fig. 2A, D, G-J; Fig. 5A, B).

Minor comments:

Introduction:

- Page 3 line 71-72: "is" is missing between "which" and "required".

Methods:

- Describe how each of the reporter constructs generated in this study was generated. Include information about the cis-regulatory sequences used.

- Describe how many sgRNAs were used to induce knockout mutation for each target gene.

- Page 6: Describe what "KY" and "KH" are.

Results:

- Page 8 line 218: What is the evidence that PNs are the only cell type bearing an axon in the papilla? Provide a reference.

- Line 220: What is "cross-plasmid transvection"? Please explain.

- Line 222: What is the evidence that KH.C4.78 is a specific marker of PNs?

- Page 9 line 245: Show evidence that L141.36 does not label PNs.

- Line 261: Which reporters label which cell types outside the papillae?

- Page 10 line 302: Provide validation that Islet or Sp6/7/8 are indeed overexpressed.

- Line 304: Describe how expansion of IC reporter expression was quantified.

- Page 11 Line 319: Show quantitative evidence that ACC reporter expression is normal in Sp6/7/8 mutants.

- Line 322: Why should Foxc>Sp6/7/8 abolish IC reporter expression? Does it repress Islet (and Foxg)?

- Line 324-325: Provide quantitative evidence that Foxc>Sp6/7/8 expands OC reporter expression.

- Page 12 line 353: Are ICs and OCs necessary for glue secretion? It seems possible to address this by perturbing the specification of both cell types using Sp6/7/8 CRISPR mutagenesis.

- Line 367-368: This sentence is grammatically confusing as written. The three spots being referred to do not downregulate Foxg etc. I suggest changing it to "… arising from the cells that are in between… the three spots and downregulate Foxg…"

- Page 13 line 402: What about the effects of POU4 knockout on ICs and basal cells? Are there any?

- Page 14 line 13: change in to is.

- 414: What is the expression pattern of Notch and Delta during papilla development?

- Line 417: Provide quantitative evidence that Foxc>SUH-DBM induces expansion of PN reporter expression.

- Page 16 line 471: Figure 7H -> 7F.

- Line 472: Figure 7I -> 7G

- Line 474: Show evidence that Villin expression is reduced specifically in the central cells of the papillae upon Islet knockout.

- Line 482: Provide representative images showing the difference in length along the apical-basal axis of ACCs between Villin CRISPR larvae and controls.

- Change Figure 7F -> 7H.

- Page 17 line 512: This implies that Sp6/7/8 CRISPR does not affect PN development, but was the effect of Sp6/7/8 CRISPR on PN development examined?

Discussion:

- Page 20 line 602: How are PNs communicating with ACCs or ICs? Do PNs innervate ACCs or ICs - any relevant data (EM etc)?

Figure 1:

- Indicate developmental stages in hours post fertilization as well, which would be helpful for readers unfamiliar with Ciona development to follow the manuscript.

Figure 2:

- Line 666: Panel B does not show Foxg reporter expression.

Figure 4:

- Line 703-705: Provide a quantitative definition for each category.

Figure 5:

- I found it a bit odd that different species were used to examine PNA-stained granules in ICs (in C. intestinalis) and OCs (in C. robusta). Any reason?

Figure 7:

- Line 769-770: The expression of Villin in Foxg+/Islet+ central papilla cells is not clear. Could Foxg and Islet expression be visualized together with Villin?

Supplemental Figure 2:

- Line 1028: Label ACCs and PNs in the figure panels.

- Line 1035: What is the evidence that Emx reporter expression occurs in ICs?

Rev. 2:

The manuscript 'Specification of distinct cell types in a sensory-adhesive organ for metamorphosis in the Ciona larva' by Johnson et al. is a beautiful piece of work that extensively describes the different cell types that compose the adhesive papilla, the molecular mechanisms regulating their specification, and their function during metamorphosis. The later aspect is yet exploratory but strongly supports the idea that metamorphosis is a complex process involving multiple cellular actors and molecular mechanisms. Importantly, the results demonstrate that this process can be tracked and manipulated. The core of the study is on the identification and molecular specification of the various papilla cell types. The understanding of papilla formation in Ciona has reached levels that have no precedent. This study opens a new field in urochordate research, and will provide data to fuel the long standing question of the homology of the papilla with anterior vertebrate structures (cranial placodes, cement gland, telencephalon) that has gained interest recently.

The present manuscript presents some weaknesses that the authors should be able to amend quite easily.

Major points:

- The Supplemental Sequence File is absent, preventing proper full evaluation.

- The number of independent experiments that have been performed is not indicated. From Fig S6, it seems replicates have not been generally done. This is a serious issue that needs clarification from the authors.

- Given the efficient perturbation (CRISPR and overexpression) and analysis tools that have been produced, it is a pity that the fate of all cell types has not been systematically examined. While I understand that basal cell formation could deserve a specific study, additional data would make the study stronger. More specifically:

- Fig 4: PNs are missing in the analysis.

- Fig 6: the role of Notch signaling pathway in the formation of ACCs and ICs has been eluded.

- The role of Sp6/7/8 using CRISPR deserves more detailed attention. According to Liu and Satou (2019), Foxg and Isl should be ectopically expressed (the U-shaped swath). Is it the case using CRISPR? In that case, there should be no PNs (not examined, same point as above), and additional ACCs (but it is not the case in Fig 4). Without PNs the metamorphosis should be inhibited, but it is pretty normal (Fig 8). Since Sp6/7/8 has a dynamic expression pattern, a possible explanation is that CRISPR experiments only display phenotypes in the collocytes, the latter/massive domain of expression of Sp6/7/8. This aspect should be explained/discussed by the authors.

- The authors have assigned cell lineage to the different cell types through indirect evidence mainly based on gene expression patterns/gene function. This is quite reasonable, and I am not asking for direct lineage tracing experiments. However, the case of PNs is questionable. The lack of PNs following MAPK inhibition leads the authors to trace PNs to the missing territory (Foxg+ then Foxg- cells). This is in contradiction with a recent report (Roure et al., 2023) that shows a similar Foxg+/Isl+ U-shaped stripe surrounded by potential Pou4+ PNs following BMP inhibition. The authors propose that these Pou4+ cells are not PNs but other neurons (RTENs). It is also very possible that MAPK inhibition blocks PNs specification/formation. The authors have potentially three ways (MAPK inhibition, BMP inhibition and Sp6/7/8 CRISPR) to produce a Foxg+/Isl+ U-shaped stripe. It would be crucial to determine in such embryos the expression of Pou4 and PN specific markers (KH.C4.78 for example).

Minor points:

- Staging is overall poorly described, mainly by indicating XX hpf at XX°C. It is necessary to indicate stages, possibly by using the staging system defined in Hotta et al. 2007.

- Line 182: mounting method is not described earlier in the Methods section.

- Lines 207-208: it would be interesting to describe with more details the novel cell type markers (protein domains, phylogenetic distribution (Ciona specific? ascidian specific?...).

- C. robusta vs C. intestinalis cis-regulatory sequences. Without the supplemental sequence file, it is difficult to appreciate the type of regions used (position in the locus, size...). Could the authors show comparisons of the respective loci with an evaluation of their conservation at the nucleotide level?

- Fig 4 and line 319: could the authors show a quantification of the effect of Sp6/7/8 CRISPR on ACCs as in Fig 4E?

- It seems that analyses of the phenotypes have been generally performed on embryos/larvae that were positive for an 'electroporation tracer' such as Foxc>H2B::mCherry. The authors should check that is explicit in all figure legends. It seems missing in Fig 4 for example.

- Could the authors comment on the number and diversity of the cell types that they have uncovered? They have clearly identified 5 cell types from a sc-RNAseq re-analysis that yields 11 clusters (Fig S1). Are there additional cell types? Any hint of differences for a given cell type, for example along the dorso-ventral or left/right axes?

- The model presented for the regulation of ACCs, ICs and OCs by Islet and Sp6/7/8 is very nice. Could the authors comment on results that do not have a perfect match. An increase of ACCs (at the expense of ICs) is expected in Sp6/7/8 CRISPR, but it isn't the case. Similarly, an increase of OCs (at the expense of ICs) is expected in Isl CRISPR, and it does not occur.

- Possibly again because of the absence of the supplemental sequence file, it is unclear which sgRNA (alone or in combination) has been used for each result that is presented. Also, when using a different sgRNA (or a combination instead of a single one) to target the same gene, the difference in phenotype penetrance/strength is not indicated. If not tested, it should be clearly stated (for example line 447).

- Fig 6H: asterisk not described in the legend.

- Cell elongation (Fig 7). The authors focus on ACCs/ICs. But, it seems clear from their data that OCs and PNs are also elongated. Furthermore, basal cells are also more elongated than 'typical epidermis' outside the papilla region. In the Isl CRISPR, cells seem similar to basal cells. This suggests, that there may be a least two levels of cell shape control, one general to the papilla organ and one specific to the 3 protrusions.

- Line 443: Fig 7C does not show the RNAseq design.

- Line 449: 'Islet conditions', what does it mean?

- Line 455: references of already validated ACC genes seem missing. These genes could be highlighted in Supplemental Table 2.

- Lines 471-472: wrong figure panels.

- Are Astl-related and Villin expressed in both ACCs and ICs?

- ACC length measurement is not fully convincing since it seems to depend largely on the orientation of the larva and the ACCs during imaging. Also the number of cells that have been measured is not indicated. The numbers of papillae, larvae and experiments need also to be described.

---

## [Decision Letter · Decision Letter 2]

23 Jan 2024

Dear Dr Stolfi,

Thank you for your patience while we considered your revised manuscript entitled "Specification of distinct cell types in a sensory-adhesive organ for metamorphosis in the Ciona larva" for publication as a Research Article at PLOS Biology. This revised version of your manuscript has been evaluated by the PLOS Biology editors, the Academic Editor and the two original reviewers.

Based on the reviews, we are likely to accept this manuscript for publication, provided you satisfactorily address the data and other policy-related requests stated below.

In addition, we would like to make a suggestion to improve the title:

"Specification of distinct cell types in a sensory-adhesive organ important for metamorphosis of tunicate larvae"

We expect to receive your revised manuscript within two weeks. 

*Published Peer Review History*

*Press*

Sincerely,

Ines

--

Ines Alvarez-Garcia, PhD

Senior Editor

PLOS Biology

Fig. 3C; Fig. 4B, D, E, F; Fig. 5E; Fig. 6B, C, I, J, K; Fig. 7A, B, F, H, I, L; Fig. 8B, C; Fig. S4A, B; Fig. S5A, B; Fig. S7B; Fig. S8B, C, E, F; Fig. S9 and Fig. S10

DATA NOT SHOWN?

Reviewer's responses

Rev. 1:

My comments have been addressed to a satisfactory degree.

Rev. 2:

The revised version has fully addressed the comments raised in the initial review.

---

## [Editor Report · Decision Letter 3]

21 Feb 2024

Dear Dr Stolfi,

Thank you for the submission of your revised Research Article entitled "Specification of distinct cell types in a sensory-adhesive organ important for metamorphosis in tunicate larvae" for publication in PLOS Biology. On behalf of my colleagues and the Academic Editor, Selene Fernandez-Valverde, I am delighted to let you know that we can in principle accept your manuscript for publication, provided you address any remaining formatting and reporting issues. These will be detailed in an email you should receive within 2-3 business days from our colleagues in the journal operations team; no action is required from you until then. Please note that we will not be able to formally accept your manuscript and schedule it for publication until you have completed any requested changes.

PRESS

Sincerely, 

Ines

--

Ines Alvarez-Garcia, PhD

Senior Editor

PLOS Biology
